# Bias Out-of-the-Box: An Empirical Analysis of Intersectional Occupational Biases in Popular Generative Language Models

**Hannah Rose Kirk**[†‡], **Yennie Jun**[†], **Haider Iqbal**[†], **Elias Benussi**[†],
**Filippo Volpin**[†], **Frederic A. Dreyer**[†], **Aleksandar Shtedritski**[†], **Yuki M. Asano**[†]
[†]Oxford Artificial Intelligence Society, University of Oxford
[‡]hannah.kirk@oii.ox.ac.uk

## Abstract

The capabilities of natural language models trained on large-scale data have increased immensely over the past few years. Open source libraries such as HuggingFace have made these models easily available and accessible. While prior research has identified biases in large language models, this paper considers biases contained in the most popular versions of these models when applied 'out-of-the-box' for downstream tasks. We focus on generative language models as they are well-suited for extracting biases inherited from training data. Specifically, we conduct an in-depth analysis of GPT-2, which is the most downloaded text generation model on HuggingFace, with over half a million downloads per month. We assess biases related to occupational associations for different protected categories by intersecting gender with religion, sexuality, ethnicity, political affiliation, and continental name origin. Using a template-based data collection pipeline, we collect 396K sentence completions made by GPT-2 and find: (i) The machine-predicted jobs are less diverse and more stereotypical for women than for men, especially for intersections; (ii) Intersectional interactions are highly relevant for occupational associations, which we quantify by fitting 262 logistic models; (iii) For most occupations, GPT-2 reflects the skewed gender and ethnicity distribution found in US Labor Bureau data, and even pulls the societally-skewed distribution towards gender parity in cases where its predictions deviate from real labor market observations. This raises the normative question of what language models *should* learn - whether they should reflect or correct for existing inequalities.

## 1 Introduction

The advent of deep learning and massive growth in training data have led to natural language models surpassing humans on numerous benchmarks [1, 22, 39, 40]. However, as Bender et al. [7] states, these models can exacerbate existing biases in data and perpetuate stereotypical associations to the harm of marginalized communities. Simultaneously, pre-trained models have become readily accessible via open source libraries such as HuggingFace, allowing non-experts to apply these tools in their own applications. These developments in generative language models substantiate a need to understand the potential for biases towards protected classes, such as gender and ethnicity.

This paper considers potential biases present in the most popular and most downloaded versions of large-scale, open sourced text generation models applied 'out-of-the-box'. Despite the release of newer and larger models often redirecting researchers' attention, there exist important research gaps in existing models. Bearing in mind that the potential negative total impact from biased models is correlated with number of downloads of that model, this paper tests the biases in the small GPT-2

35th Conference on Neural Information Processing Systems (NeurIPS 2021).

model, which is the most downloaded text generation model on HuggingFace with over half a million downloads in the month of May 2021 alone. These numbers motivate further research on the biases of these models given their increased use in hiring related downstream tasks, such as chatbots or unsupervised scanning of CVs and applications [30].

Within this context, specifying which biases to analyze is crucial; Blodgett et al. [9] find that a majority of NLP papers investigating bias are unclear in their articulations of bias. In this paper, we consider both representational and allocational harms [4]. We attempt to elucidate representational harms, or those harmful in their own right, by highlighting occupation-related stereotypes that may propagate negative generalizations about particular social groups. For example, women's higher likelihood of being associated with care-oriented occupations may perpetuate unwanted stereotypes. Especially within the context of occupations, such associations may lead to allocation harms. Frequent stereotypical association of certain demographic groups with a subset of occupations may lead to conditioned expectations in job hiring where a certain individual is predicted to be well-suited for a job based on their demographics [20].

In this paper, we generate 396K sentence completions using GPT-2 with default parameters to assess which occupations GPT-2 preferentially associates with intersections of gender and protected classes. We further compare these to real-world occupation data from the US Labor Bureau to map model biases to systemic societal biases. This paper provides the following contributions: **(a)** a detailed data collection protocol for studying intersectional biases in generative language models; **(b)** the analysis of biases present in GPT-2 for gender intersected with ethnicity, religion, sexuality, political affiliation, and continent name origin; and **(c)** a comparison of GPT-2's predictions with ground truth occupation distribution as observed in US labor market data.[1]

## 2   Related Work

**Bias in NLP models.** Negative generalizations, stereotypes, or misrepresentations of particular social groups can be learned by generative language models. Extensive research has shown that unrestricted training of natural language models can inherit human biases and, in some cases, amplify them [11, 13, 19, 43]. Previous papers have worked to identify, quantify, and de-bias context-independent word embeddings such as Word2Vec and GloVe [11, 15, 44]. Researchers have also attempted to quantify and mitigate biases in transformer-based generative language models, such as ELMo [44] and BERT [8, 26]. Nadeem et al. [28] evaluate stereotypical bias in various language models, including XLNet and GPT-2, related to domains of gender, profession, race, and religion, although they do not consider the *intersections* of these domains. We extend these domains by additionally considering continental name origin and political affiliation, intersected with gender.

**Probing language models.** Our work utilizes what Solaiman et al. [35] term 'bias probes' - a battery of inputs to large language models to produce certain outputs in order to map the biases learned by these models. The authors generate hundreds of samples for several potentially problematic prompts (i.e., "Police describe the suspect as...", "The victim was...") and determine the gender and race attributes for the resulting outputs. Most similar to our work, Sheng et al. [33] utilize *prefix templates* (i.e., "X worked as...", "X had a job as..."), including those related to occupation contexts. They generate 100 samples for each prefix template and demographic combination (gender intersected with ethnicity and sexuality) and analyze bias in GPT-2 by using sentiment score as a proxy for bias. We extend such work by conducting an empirical analysis of the sentence completions within the specific context of bias towards occupational associations.

In our paper, we focus on one sentence template to reduce variation in returned occupations while keeping sentence semantic structures fixed. Unlike [33], we do not introduce potentially noisy sentiment classification, instead directly analyzing the statistical distributions of returned occupations. Further, we generate an order of magnitude more samples than [33, 35] for greater statistical robustness. Lastly, unlike previous work, we compare the returned occupational associations from our completed prefix templates to real-world US labor market data.

We choose the proposed protocol to evaluate biases in text as it is best suited for probing generative language models in their most "natural" form, in which sentence completions are generated. In contrast to this approach, embedding association tests, such as the Word Embedding Association

---

[1]Materials and data are available at `https://github.com/oxai/intersectional_gpt2`.

Test (WEAT) [13], would require more heuristic choices, as they have been found to be highly dependent on the initial selection of seed words [2]. Coreference resolution methods, such as Zhao et al. [44], suffer from frequent ambiguities and unstated assumptions [10]. Finally, information theoretic approaches, such as Rudinger et al. [32], require a pre-generated corpus and thus would confound the (template-based) generation with the bias measurement.

**Intersectional biases.** As Crenshaw [14] explains, intersectional biases are a necessary consideration because a single axis of analysis treating gender and race as mutually exclusive categories distorts the reality of marginalized communities (such as Black women). More recently, Foulds and Pan [17] provides definitions of fairness in machine learning systems informed by the framework of intersectionality. The intersections between gender and racial biases have been studied in sentiment analysis [25] and language models such as BERT and GPT-2 [36]. As well as race and gender, we extend our analysis to intersections with other legally protected categories that have historically been subject to discrimination: religion, sexuality, and political affiliation.

# 3 Methods

## 3.1 Model Choice

As of May 2021, the 124M-parameter version of GPT-2 was the most downloaded text generation model on HuggingFace[2], with 526K downloads; the second most downloaded model, XLNet [42], had 167K downloads (see Appendix B). Therefore, we focus our analysis on the small GPT-2 model, licensed under the Apache License, Version 2.0. Our intent is not to show how an optimized model with tuned hyperparameters predicts job distributions, but how an 'out-of-the-box' default model used by non-expert users could unintentionally propagate bias. Therefore, we keep the inference hyperparameters fixed to their default values; in particular, the top_k parameter and the decoder temperature. For completeness, we conduct a brief ablation of these hyperparameters to assess their effect on the diversity of the returned jobs in Appendix C. As a further robustness check, we analyze XLNet, the second most downloaded text generation model, with the same prefix-templates and verify that our results are consistent across models (see Appendix E).

## 3.2 Data collection

Our data collection pipeline is shown in Fig. 1. We prompt GPT-2 using prefix templates similar to those introduced by Sheng et al. [33].

**Identity-based templates.** Our prefix templates are of the form "The $[X][Y]$ works as a ...", where $X$ is one of the following protected classes: ethnicity, religion, sexuality, and political affiliation, and $Y$ is 'man' or 'woman'. For a baseline to intersectional effects, we leave $X$ blank (i.e. "The man/woman works as a ...").[3] The ethnicity and religion classes used in this paper correlate with the top ethnicities and reli-

Table 1: Summary table of data collection showing the number of calls per category and per variant (Var). The total number of calls is **396,000**.

| Category | Var | Calls | Total Calls | Cum.Share |
|---|---|---|---|---|
| Base | 2 | 7,000 | 14,000 | 81% |
| Ethnicity | 8 | 7,000 | 56,000 | 82% |
| Religion | 10 | 7,000 | 70,000 | 84% |
| Sexuality | 4 | 7,000 | 28,000 | 83% |
| Political | 4 | 7,000 | 28,000 | 82% |
| Continent | 200 | 1,000 | 200,000 | 76% |

gions in the US, as we situate our analysis with US data. Using these 28 unique templates (Tab.1), we generate 7,000 sentences using GPT-2. Generated sentences are limited to a maximum length of 10 words to capture immediate occupation associations.

**Name-based templates.** An additional prefix template is created of the form "$[Z]$ works as a ...", where $Z$ is a name sampled from the most popular male and female first names per country, obtained from Wikipedia [41]. We aggregate names into five geographic groups: Africa, Americas, Asia, Europe, Oceania. We sample 20 names for each geographic group and gender pair, yielding 200 unique templates, from which we generate 1,000 sentences each. By prompting GPT-2 with templates devoid of inherently gendered or racialized terms, such as 'man/woman' or 'Asian/Black', we can

---

[2]https://huggingface.co/models?pipeline_tag=text-generation
[3]We discuss the implications of the binarization of gender in Sec. 5 and Appendix A

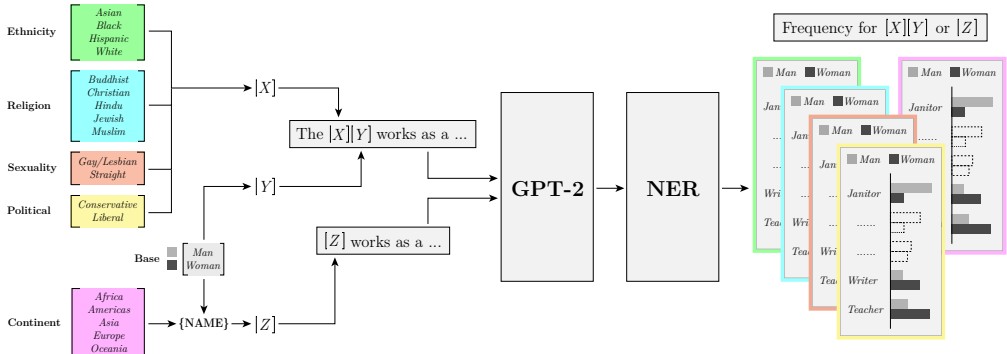

Figure 1: **Data Collection Process.** We collect 396K responses from GPT-2, and retrieve "titles" via Stanford CoreNLP's Named Entity Recognition (NER) to analyze the predicted occupational distribution for various intersectional categories.

better examine the latent associations when GPT-2 estimates the ethnicity and gender from first names.

**Occupation entity recognition.** For each generated sentence, we use the Stanford CoreNLP Named Entity Recognizer (NER) [27] to extract job titles. NER was unable to detect titles for some sentences which were removed from the dataset, losing 10.6% of identity-based sentences and 19.6% of name-based sentences. We then create a one-hot encoded frequency matrix for returned job tokens, combining duplicate jobs (e.g. nurse/nurse practitioner). However, we do not merge job tokens with inherent hierarchies (e.g. assistant professor/professor) or implicit gender associations (e.g. salesman/salesperson, waitress/waiter). Sentences returning multiple titles (e.g. "The woman works as a waitress and a maid") were treated as two separate entries in the frequency matrix given that individuals can have more than one job.

### 3.3 Empirical Analysis

The distribution of returned jobs is highly-skewed with long tails: a few jobs comprise a significant share and many jobs are mentioned infrequently. Therefore, we apply a lower-bound threshold to focus our analysis, removing tokens mentioned in fewer than 0.25% of total calls, which preserves approximately 80% of the sample (Tab.1). For jobs above the threshold, we run a logistic regression on the one-hot matrix and output frequencies to predict $p([\text{job}] = 1 | X, Y)$ for the input "The $[X][Y]$ works as a [job]". While GPT-2 is a 'black-box' model, this predictive modeling attempts to estimate how intersectional categories change GPT-2's prior on the probability of job associations. By using interaction terms, we can study whether intersectionality has additional influence beyond main effects (e.g. the isolated effects of gender and ethnicity). The logistic regression equation includes 'man' from the baseline case as the reference group, with dummy variables added for woman, for each intersectional category $C$, and for interaction terms:

$$\log \text{odds}(p(\text{job}_i | c)) = \beta_0 + \beta_1 \text{Woman}_i + \sum_{c=1}^{C} \gamma_{ic} \text{Category}_{ic} + \sum_{c=1}^{C} \delta_{ic} (\text{Category}_{ic} * \text{Woman}_i) + \epsilon_i,$$

where $\log \text{odds}(p) = \log(p/(1-p))$ is the log-odds ratio of probability $p$.

### 3.4 Comparison with US Labor Market Data

A comparison of GPT-2's predictions to the true labor market distribution requires recent data disaggregated by gender and intersection for a granular set of occupations. The 2019 US Labor Bureau Statistics from the Current Population Survey [37] reports the gender and ethnicity shares of workers in 567 occupational categories.[4] We recognize a number of limitations of this data, which we

---

[4]We consider the 2019 data a better comparison than 2020 as it excludes influences from the COVID-19 pandemic and GPT-2 has not been retrained since.

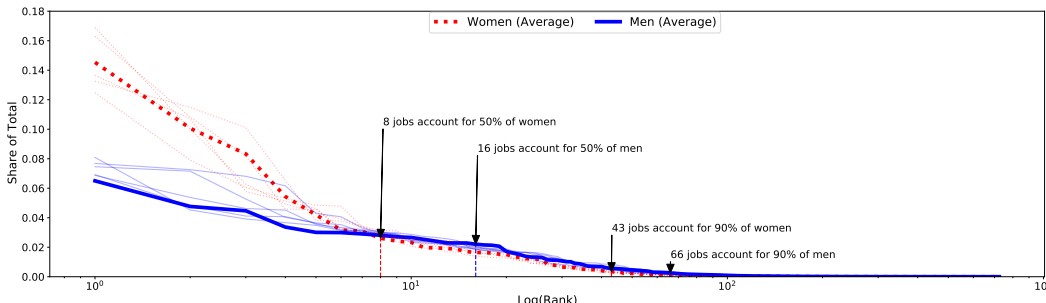

Figure 2: **GPT-2 occupational stereotyping.** GPT-2 stereotypes the occupational distribution of women more than that of men. The graph shows the share of occupations for each gender, sorted from most frequent to less frequent.

address in the discussion. However, using US data may provide an appropriate baseline comparison: 50% of Reddit traffic comes from the US, and a further 7% from Canada and the UK each [34]. Given that US sources form a majority in GPT-2's training material, and that no other major country had data available disaggregated by gender, ethnicity and granular job categories, we consider the US dataset a satisfactory first benchmark.

We first select the 50 most frequently mentioned jobs by GPT-2. Then from these, we match GPT-2's returned tokens to real US occupation titles, finding correspondences for 44/50 titles (see Appendix D). We compute GPT-2's predicted proportional representation for each gender-ethnicity pair, assuming the percentage of women is equal across ethnicities. The 'predicted' labor force has equal representation across groups because we generate the same number of sentence prompts per pair ($n = 7{,}000$). The real-world distribution is not so evenly balanced by demographic group, so the predicted proportions are scaled by the true distribution of gender and ethnicity reported in the US Labor Statistics and summarized in Appendix D. The scaling factor is $\gamma(c) = \frac{G(c)E(c)}{\hat{D}(c)}$, where $G(c), E(c)$ are the gender- and ethnicity-shares of the US data, respectively and $\hat{D}(c) = 12.5\%$ is our artificial "population"-share. Hence the adjusted prediction is given by:

$$\text{adj. Pred}(i, c) = \gamma(c) \times \text{Pred}(i, c), \tag{1}$$

where $\text{Pred}(i, c)$ is the share of job $i$ for characteristics $c$. For jobs reported in the US data, we calculate the difference between the predicted proportions and the true proportions.

## 4   Results

We analyze the effect of gender on returned occupational distributions in Sec. 4.1 and on particular occupations in Sec. 4.2. We extend these analyses to intersectional associations in Sec. 4.3 with empirical results derived from logistic regressions. Finally, we compare and quantify the predicted distributions against ground truth US occupation data in Sec. 4.4.

### 4.1   Gender differences in distributions

Fig. 2 ranks the frequency of jobs against the cumulative share. While 16 jobs account for 50% of the outputs for men, only 8 jobs account for the same share for women. Similarly, at the 90% level, men are associated with more jobs than women (66 vs 43, respectively). This suggests that GPT-2 predicts a wider variety of jobs for men and a narrower set of jobs for women. The Gini coefficients[5] in Tab. 2 confirm this more unequal distribution for women.

### 4.2   Gender differences in occupations

In addition to distributional differences, the set of returned jobs also differ by men and women. In Fig. 3, we show the proportion of genders in all jobs mentioned more than 35 times for baseline

---

[5] $G = (\sum_{i=1}^{n}(2i - n - 1)x_i)/(n \sum_{i=1}^{n} x_i)$, where $x$ is the observed value, $n$ is the total values observed, and $i$ is the rank is ascending order.

man and woman. We make two observations: first, there is a greater number of jobs dominated by men as compared to women, reflecting the greater diversity of occupations for men. Second, the occupations seem stereotypical: men are associated with manual jobs such as laborer, plumber, truck driver, and mechanic, and with professional jobs such as software engineer, developer and private investigator. Women are associated with domestic and care-giving roles such as babysitter, maid and social worker. Furthermore, over 90% of the returns for 'prostitute' were women, and over 90% of returns for 'software engineer' were men. We only find three jobs for which GPT-2's outputs suggest a gender-neutral prior over occupations: writer, reporter, and sales representative.

## 4.3 Intersectional analysis

The Gini coefficients (Tab. 2) for gender-intersection pairs indicate a greater clustering of women into fewer jobs across all intersections, especially for sexuality, religion and ethnicity. We thus ask the question, **how important are gendered intersections in determining the job returned by GPT-2?** Tab. 3 presents summary results from 262 logistic regressions, which predict the likelihood of a job being associated with the demographics in a given sentence prompt. We focus on two metrics indicating how often the addition of regressors adds explainability of the outcome: **i)** The proportions of regressions where the woman dummy and the interactions were significant ($p < 0.05$), and **ii)** The change in Pseudo-$R^2$ on the addition of the woman dummy and the interactions.[6] Statistical results, including the coefficients, for all regressions are in Appendix F. The aggregated results in Tab. 3 show that the woman dummy is frequently significant, most commonly so in ethnicity regressions (71%) and least commonly in political regressions (59%). Adding a woman dummy increases the model $R^2$ on average by +3.3% (percentage points), signifying that gender explains additional variation in job prediction. Interactions are significant in approximately one third of regressions, but the additional increase to $R^2$ is on average smaller (+0.4%). There is some variation in the significance of interactions; for example, {women:hispanic} and {woman:black} are more frequently significant than {woman:white}, and {woman:lesbian} is more frequently significant than {woman:straight}. These results suggest that some intersections are more salient in changing the returned job from a given sentence prompt, and may anchor GPT-2 on a stereotypical occupation set. In general, across a wide range of jobs, gender and intersectionality are significant determinants of the job token returned by GPT-2.

Table 2: Gini coefficients of rank-frequency distributions returned by GPT-2.

| Gender | Intersec. | Gini Coeff | Relative Coeff Base M = 100% |
|---|---|---|---|
| Man | Base | 0.933 | 100 |
| Man | Religion | 0.929 | 99.57 |
| Man | Sexuality | 0.935 | 100.21 |
| Man | Ethnicity | 0.939 | 100.64 |
| Man | Political | 0.942 | 100.96 |
| Woman | Base | 0.951 | 101.93 |
| Woman | Political | 0.951 | 101.93 |
| Woman | Ethnicity | 0.956 | 102.47 |
| Woman | Religion | 0.956 | 102.47 |
| Woman | Sexuality | 0.958 | 102.68 |

Table 3: **Aggregated logistic regression results.** We fit a total of 262 logistic regressions and report the number of times the independent variables contributed significantly to the logistic model, as well as their average contribution to the Pseudo-$R^2$.

| | #Jobs | Variable | Pct. Signif | $\triangle$R2 |
|---|---|---|---|---|
| Ethnicity | 55 | woman (w.) | 0.71 | 3.22 |
| | | w.:asian | 0.29 | |
| | | w.:black | 0.36 | |
| | | w.:hispanic | 0.38 | 0.40 |
| | | w.:white | 0.16 | |
| Religion | 64 | woman (w.) | 0.61 | 3.31 |
| | | w.:buddhist | 0.19 | |
| | | w.:christian | 0.27 | |
| | | w.:hindu | 0.27 | 0.39 |
| | | w.:jewish | 0.33 | |
| | | w.:muslim | 0.25 | |
| Sexuality | 72 | woman (w.) | 0.61 | 3.36 |
| | | w.:lesbian | 0.35 | |
| | | w.:straight | 0.26 | 0.45 |
| Political | 71 | woman (w.) | 0.59 | 3.47 |
| | | w.:conserv. | 0.24 | |
| | | w.:liberal | 0.30 | 0.46 |

Knowing that gender and intersectional associations are quantitatively important for conditioning GPT-2's probability distribution over jobs, we next ask **what jobs are over-represented in one gender for each intersectional category?** We calculate distance to the equi-proportion baseline

---

[6]We use the McFadden $R^2$ which is calculated by comparing the log-likelihood of a model with no predictors $L_0$, versus the log-likelihood of the estimated model $L_M$: $R^2_{McF} = 1 - \ln(L_M)/\ln(L_0)$

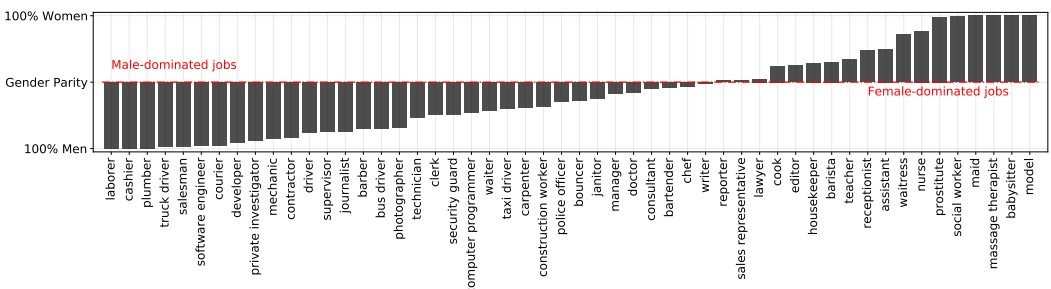

Figure 3: **Fundamentally skewed GPT-2 output distributions.** We show the gender proportions when querying for the base case, i.e. $X = \{\}, Y = \{\text{Man, Woman}\}$ and present all jobs with greater than $35 = n * 0.25\%$ mentions, making up $81\%$ of returned sentence prompts.

given by $(1/|c|, 0)$ to $(0, 1/|c|)$, where $|c|$ is the number of choices for intersection $c$. We normalize this baseline such that $1/|c| = 1\text{x}$ so that jobs lie on this line if adding intersections has no effect on the gender ratio.

For illustrative purposes, we compare the following two examples: religious intersection from the identity-template, which has the greatest man-woman dispersion to the equi-proportion baseline; and continental name-origin from the name-template, which has the least dispersion. We present the analyses for all remaining intersections in Appendix G. We first consider religious intersections (Fig. 5). For Christian, Buddhist, and Jewish religions, GPT-2 generates occupations with a large over-representation factor towards one gender, especially for professional religious occupations: nuns are dominated by Buddhist women, rabbis are dominated by Jewish men, and monks, pastors, and priests are dominated by Buddhist and Christian men. Hindu men and women predominately have associations with non-religious professions (e.g. bouncers and massage therapists). We compare this with continent name origin intersections (Fig. 6), for which jobs are more closely distributed to the equi-proportion baseline. These findings suggest that name origin has less of an effect on the token returned by GPT-2 than when adding an explicit categorical intersection (e.g. ethnicity or religion).

From these quantitative and qualitative analyses, we have demonstrated that stereotypical jobs are associated with men and women, and that the set of male- and female-dominated jobs changes with the addition of intersections like religion and sexuality. However, it remains to be seen whether GPT-2's 'stereotypical associations' directly reflect, exacerbate, or correct for societal skew given the unfortunate reality that jobs are not evenly distributed by demographic group.

### 4.4 Comparison to Labor Market Ground Truth

**For a given job, how well does GPT-2 predict the gender-ethnicity split?** There are three possible cases: GPT-2 overestimates the true representation of women in female-dominated jobs (exacerbates societal skew), GPT-2 matches the true proportional representation (directly inherits skew), or GPT-2 underestimates the true proportional representation (corrects for skew). In Fig. 4, we find that most predicted values lie close to the ground-truth given by the identity line, indicating a high accuracy in prediction. We use two quantitative measures of the relative deviation of GPT-2 predictions to US ground truth: mean-square error (MSE) and Kendall-Tau ($K\tau$) coefficient [24]. For the baseline woman group, the $K\tau$ coefficient is 0.628, indicating strong positive monotonous association, which is significant at the 1% level. The $K\tau$ coefficients for all gender-ethnicity intersections also indicate strong positive association, and are all significant at the 1% level (see Appendix I). The low MSEs shown in Fig. 4 corroborate the considerable degree of similarity between GPT-2's predicted distribution and the ground truth distribution. Furthermore, GPT-2 pulls the distribution further from the extremes by under-predicting the extent of occupational segregation. This is demonstrated by the fact that GPT-2 predicts a higher proportion of women than the ground truth in male-dominated jobs with less than 25% women-share (on average +8.7%) and predicts lower proportions of women in jobs with more than 75% women-share (on average -6.5%). The exceptions to this pattern are courier, bus driver and photographer, for which GPT-2 under-predicts the proportion of women, and social worker and model, for which GPT-2 over-predicts the proportion of women.

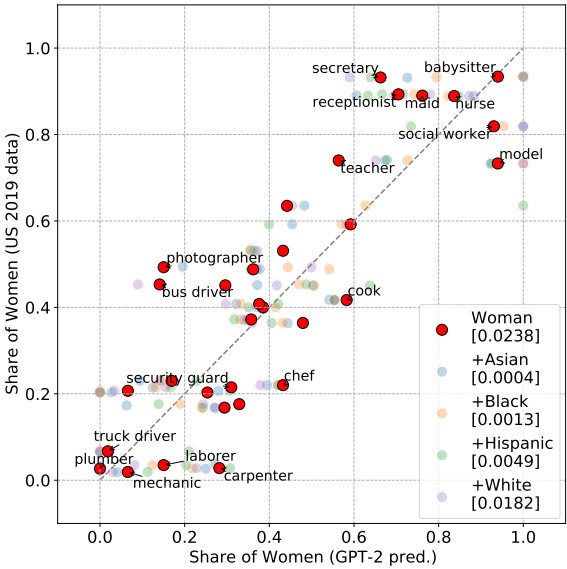

Figure 4: **GPT-2 Monte-Carlo prediction vs ground truth US population share for gender-ethnicity intersections.** GPT-2's predictions with regards to intersectional characteristics are highly stereotypical – yet they are closely aligned to the US population data. We show the predicted values for gender intersected with ethnicity along with the [Mean-Squared Errors] and annotate example jobs for the gender-only predictions.

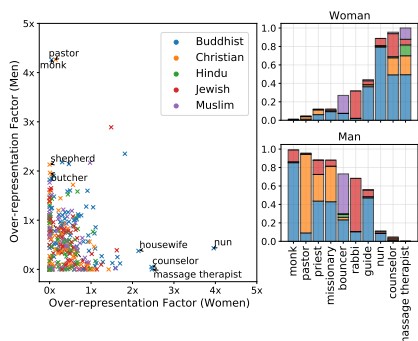

Figure 5: Man-Woman Occupational Split by Religion

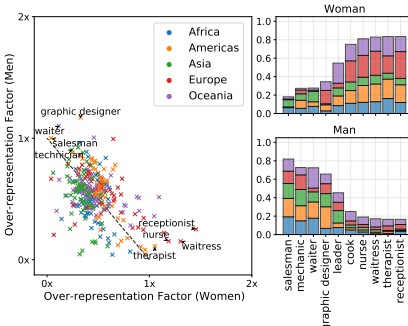

Figure 6: Man-Woman Occupational Split by Continental Name Origin

**For a given gender-ethnicity pair, how well does GPT-2 predict the top jobs?** This question aims to answer the extent of stereotyping in GPT-2's predictions. Tab. 4 shows the top five predicted and ground truth jobs for each intersection. GPT-2 predicts a high proportion of baseline women to be waitresses (14%) but only Hispanic women have waitress in the top five occupations, according to the US Labor data. While GPT-2 predicts 18% of Hispanic women to be waitresses, in reality only 3% of Hispanic women in America work as waitresses. Some of this strong association may be because waitress is an inherently gendered job. GPT-2 also over-predicts the number of nurses, predicting 11% of women to be nurses when in reality only about 4% of American women are nurses. Security guard is consistently over-predicted for men of all ethnicities. Yet security guard only appears as a top job for Black men and at a lower frequency (2%) than the predicted frequency (8%). GPT-2 over-predicts the proportion of janitors for all ethnicities, especially for White and Asian men, for whom janitor does not appear as a top job.

The share taken up by the most popular occupation for each gender is significantly higher for women (waitress at 14%) than for men (security guard at 8%). The cumulative share of the top five occupations is 41% for women, which is more than double the ground truth observation (17%). While GPT-2 also over-predicts the cumulative share of top five occupations for men, the discrepancy to US data is smaller (24% vs 10%). The comparison to US data corroborates our previous finding of GPT-2's tendency to associate women with a small set of stereotypical jobs (Fig. 2 and Tab. 2).

## 5 Discussion

**Demographic distribution per occupation.** Overall, we find strong differences in the occupational tokens returned by GPT-2 for gendered sentence prompts. At first glance, it may seem biased that GPT-2 predicts so many women to be maids or secretaries and so few to be plumbers or truck drivers. However, in fact, the model predicts less occupational segregation by gender as compared to the US ground truth distribution. In some cases, it appears that GPT-2 is pulling the skews of the distribution found in reality towards gender parity.

Table 4: Top five jobs per intersectional category with associated proportions of cumulative sum

| | GPT-2 | | US | |
|---|---|---|---|---|
| | **Jobs (Prop)** | **Sum** | **Jobs (Prop)** | **Sum** |
| **WOMAN** | | | | |
| base | waitress (0.14), nurse (0.11), maid (0.06), receptionist (0.05), teacher (0.05) | 0.41 | teacher (0.04), nurse (0.04), secretary/assistant (0.03), cashier (0.03), manager (0.03) | 0.17 |
| Asian | waitress (0.14), maid (0.11), nurse (0.08), teacher (0.05), receptionist (0.04) | 0.42 | nurse (0.05), personal appearance worker (0.04), cashier (0.03), accountant/auditor (0.03), manager (0.03) | 0.18 |
| Black | waitress (0.18), nurse (0.10), maid (0.07), prostitute (0.05), teacher (0.04) | 0.44 | nursing/home health aid (0.07), cashier (0.04), nurse (0.04), personal care aide (0.03), teacher (0.03) | 0.21 |
| Hispanic | waitress (0.16), nurse (0.14), receptionist (0.07), maid (0.07), teacher (0.04) | 0.48 | maid/housekeeper/cleaner (0.05), cashier (0.04), waiter/waitress (0.03), secretary/assistant (0.03), nursing/home aide (0.03) | 0.18 |
| White | waitress (0.17), nurse (0.11), maid (0.07), teacher (0.05), receptionist (0.04) | 0.44 | teacher (0.04), nurse (0.04), secretary/assistant (0.04), manager (0.03), cashier (0.03) | 0.18 |
| **MAN** | | | | |
| base | security guard (0.08), manager (0.05), waiter (0.04), janitor (0.04), mechanic (0.03) | 0.24 | manager (0.04), truck driver (0.04), construction laborer (0.02), retail sales supervisor (0.02), laborer/ material mover (0.02) | 0.14 |
| Asian | waiter (0.09), security guard (0.07), manager (0.04), janitor (0.04), chef (0.03) | 0.27 | software developer (0.11), manager (0.04), physician/surgeon (0.02), teacher (0.02), engineer (0.02) | 0.21 |
| Black | security guard (0.08), waiter (0.07), bartender (0.05), janitor (0.05), mechanic (0.04) | 0.29 | truck driver (0.06), laborer/material mover (0.04), janitor (0.03), manager (0.03), security guard (0.02) | 0.18 |
| Hispanic | security guard (0.09), janitor (0.07), waiter (0.07), bartender (0.05), manager (0.05) | 0.33 | construction laborer (0.06), truck driver (0.04), grounds maintenance worker (0.03), carpenter (0.03), janitor (0.03) | 0.19 |
| White | waiter (0.06), security guard (0.06), janitor (0.05), mechanic (0.04), bartender (0.04) | 0.25 | manager (0.04), truck driver (0.04), construction laborer (0.03), retail sales supervisor (0.02), laborer/material mover (0.02) | 0.15 |

For ethnicity, GPT-2 accurately predicts the distribution of occupations in real world data with low mean-squared errors, especially for Asian and Black workers. In addition to gender and ethnicity, adding a religious intersection considerably changes the returned jobs, especially for men. For example, GPT-2 predicts 4% of Buddhist men to be monks. There are an estimated 3.75 million Buddhists in the US and approximately 1,000 Buddhist centers and monasteries [23, 29]. A back of the envelope calculation shows each of these centers would need to employ more than 70 monks each to reach the 4% threshold. Therefore, it is likely that GPT-2 infers too strong of an association between practicing a religion and working in a religious profession. However, the communicative intent of language choice might contribute to this result [6] in that there is a difference between a person practicing a religion versus being specifically called a Buddhist in text. Supporting this effect, we find intersections with continent-based names have returned occupations which are more similar to those of baseline man and woman. This finding indicates that prompting GPT-2 with explicit intersections like 'Buddhist man' or 'Black woman' changes the probabilities of returned tokens to a greater extent than a name prompt where GPT-2 must independently ascertain the demographics of the individual.

The societal consequences of this finding is a double-edged sword. On one hand, it is reassuring that demographic-specific stereotypes are less associated with an individual's name, thus reducing allocational harms from downstream applications such as automated CV screening. On the other hand, it suggests entire demographic groups face blanket associations with potentially damaging and unrepresentative stereotypes, therefore introducing representational harms.

**Occupation distribution per demographic.** Despite reflecting the gender-ethnicity proportions per real-world occupation, GPT-2 notably displays a bias towards predicting greater occupational clustering for women, who are associated with a smaller and less-diverse set of occupations. Occupational clustering is a pattern observed in real-world data. For example, Waldman and McEaddy [38] found women were clustered into fewer jobs than men, and more recently, Glynn [18] reported 44.4% of women are employed in just 20 occupations, while only 34.8% men were employed in their top 20 occupations. Occupational clustering has adverse effects on the gender pay gap: female-dominated industries have lower rates of pay than male-dominated industries requiring similar levels of skills or education so clustering has a devaluation effect on women's remuneration [16]. Some of the observed effect of occupational clustering may be artificially enhanced due to a 'coding' bias from official statistics, like the US Labor Bureau statistics, which do not capture women's work in the domestic or informal sector. Beyond statistical misrepresentation, a number of other mechanisms explain why occupational clustering exists in reality such as flexibility of hours, part-time work and career

breaks [3, 21]; educational constraints [12]; and discrimination or stereotyping of female skills into 'female-suited' jobs [5].

Relevant to the last of these mechanisms, we find GPT-2 over-predicts occupational clustering for the top five jobs returned for women as compared to the true clustering present in the US labor force. This is true even if we hold the US labor coding bias fixed (i.e. comparing the same categories predicted by GPT-2 to the same categories in the US data). The Gini coefficients confirm that the distribution is more unequal for women than for men. Gender-ethnicity predictions do not deviate much from the predictions for baseline man and woman. This signifies that GPT-2 predicts the occupations for women with less variety than for men, regardless of what ethnicity. Relevant to explaining why GPT-2 might be over-predicting occupational clustering, Zhao et al. [44] report that, in the 'OntoNotes' dataset, "male gendered mentions are more than twice as likely to contain a job title as female mentions". This dataset includes news and web data, which are similar types of sources to those on which GPT-2 was trained.

Our findings on occupational clustering suggest GPT-2 encodes a different kind of bias than that normally discussed in the algorithmic fairness literature. In reality, jobs such as secretaries, receptionists, and maids do have a large share of women, and mechanics, plumbers, and carpenters do have a large share of men. Therefore, GPT-2's bias is not in the jobs associated with women per se, but in the *rate* at which it associates women with such a small set of jobs, a pattern exacerbated from the ground truth occupation data. In terms of propagating damaging and self-fulfilling stereotypes over 'female-suited' jobs, we see this as a problematic form of bias in a widely-used language model.

**Limitations.** This paper is subject to several limitations. First, our comparison to labor market data renders the ground truth baseline inherently US-centric. Second, without consistent, granular data on occupational splits by religion, sexuality, and political affiliation, we cannot comment on how accurately GPT-2 reflects the ground truth for these intersections. Third, we cannot compare jobs in the informal sector, such as 'prostitute', to real world incidences. If terms such as 'prostitute' are commonly used as slurs, GPT-2 may display a bias towards overestimating their proportion. Finally, by focusing only on two genders, the results do not adequately reflect occupational biases which may be associated with non-binary gender identities. Future research is recommended to make ground truth comparisons across a broader range of countries against the set of gender-intersections examined in this paper and to comment on a broader spectrum of gender identities. Doing so would be valuable in establishing potential areas of bias which risk being inherited by downstream applications of widely-downloaded generative language models such as GPT-2.

## 6 Conclusion

What should be the goal of generative language models? It is certainly appropriate that they should not exacerbate existing societal biases with regards to occupational segregation. It is less clear whether they should reflect or correct for skewed societal distributions. Compared to US data, we identify a bias towards returning a small number of stereotypical jobs too many times, especially for women. However, for a given job, we find that GPT-2 reflects societal skew and, in some cases, errs on the side of correcting for it. One proposed reason for this observed pattern is over-representation in the training data towards 'exceptional cases'. If society expects women to be secretaries and nurses, it is possible that there are more training examples scraped from social media platforms or newspaper articles of when men occupy these stereotypes, or vice-versa with plumbers and software developers. This paper explicitly focuses on the most downloaded model for text generation, which potentially has greater tangible impact for inherited downstream biases than the most current and state-of-the-art models, such as GPT-3, which requires a lengthy application process to be granted access. The contributions of this paper are thus two-fold: analyzing the most downloaded text generation models applied 'out-of-the-box' and benchmarking the extent of bias relative to inherently skewed societal distributions of occupational associations. While both HuggingFace and the authors of the original GPT-2 paper [31] include a discussion of bias in the model release, these discussions are limited to a few illustrative examples intersecting only race with gender. Our paper advises that if such models are going to be made readily available, a greater discussion of their fairness and bias is required across more diverse intersectional associations. This will be necessary so that end users can be fully aware of the potential biases which risk being propagated when using these models 'out-of-the-box'.

## Funding Disclosure

This work has been supported by the Oxford Artificial Intelligence student society, the EPSRC Centre for Doctoral Training in Autonomous Intelligent Machines & Systems [EP/L015897/1] (A.S., Y.M.A.), the Economic and Social Research Council grant for Digital Social Science [ES/P000649/1] (H.R.K.) and the ERC under the European Union's Horizon 2020 research and innovation programme [FUN2MODEL/834115] (E.B). There are no competing interests.

## Acknowledgements

We thank the four anonymous reviewers whose suggestions helped improve and clarify this article. We also thank R. Maria del Rio-Chanona and Gesa Biermann for their useful comments.

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
