## Supplementary Appendix

## A  Note on language used in this paper

In our paper, we focus on the occupational associations with binary gender identities i.e. "man" and "woman". While we do sometimes refer to jobs dominated by women as 'female-dominated jobs', we do not make an explicit comparison to sex, i.e. prompting GPT-2 with the 'female worker is a...'. We feel strongly about the importance in studying non-binary gender and in ensuring the field of machine learning and AI does not diminish the visibility of non-binary gender identities. In future work, we hope to extend our analysis with the same data collection pipeline. For example, *womxn* is a term used in the intersectional feminist community to be inclusive of transgender woman and non-binary individuals. The sentences returned when prompting GPT-2 with 'womxn' are primarily of two types: (i) stereotypical job associations e.g. 'drag queen', 'feminist', 'crossdresser' or 'nurse', and (ii) not recognizing 'womxn' as a person noun e.g. 'The womxn works as a kind of a noodle shop', 'The womxn works as a battery', 'The womxn works as a mauve-wool hat' or 'The womxn works as a kind of virtual sex toy'. These preliminary findings suggest it is critical for future work to study occupational biases with non-binary gender identities in generative language models.

## B  GPT-2 Model Downloads

We select the most downloaded version of GPT-2 available on HuggingFace as a proxy for popularity in use-cases by experts and non-experts alike. Tab. 5 shows that the small version of GPT-2 has an order of magnitude more downloads as compared to the large and XL versions. While using the small version of GPT-2 limits the number of hyperparameters, there are some benefits. Larger models of GPT-2 have been shown to have an increased capability to memorize training information, introducing privacy concerns [2]. Further, while the environment cost of inference is cheap, Bender et al. [1] discuss how the environmental impact of training scales with model size, and the associated consequences likely disproportionately affect marginalized populations. In Tab. 6, we show the top ten downloaded text generation models on HuggingFace, which governed our choice for selecting GPT-2.

Table 5: GPT-2 models available on HuggingFace by number by total downloads as of May 23, 2021

| Model | # Hyperparameters | # Public Downloads |
|---|---|---|
| GPT-2 Small | 124M | 526k |
| GPT-2 Medium | 355M | 140k |
| GPT-2 Large | 774M | 52k |
| GPT-2 XL | 1.5B | 31k |

Table 6: Top 10 downloaded models from HuggingFace as of May 23, 2021.

| Model Name | # Public Downloads |
|---|---|
| gpt2 | 526k |
| xlnet-base-case | 167k |
| gpt2-medium | 140k |
| chirag2706/gpt2_code_generation_model | 111k |
| EleutherAI/gpt-neo-1.3B | 109k |
| distilgpt2 | 95k |
| EleutherAI/gpt-neo-2.7B | 89k |
| gpt2-large | 52k |
| sshleifer/tiny-ctrl | 43k |
| sshleifer/tiny-gpt2 | 37k |

# C GPT-2 Hyperparameter Ablation

What is the effect of changing the default hyperparameters on the diversity of returned jobs? We focus on two of the default hyperparameters: top k, which determines the number of highest probability vocabulary tokens to keep in token generation (default = 50); and `temperature`, which modulates the next token probabilities used in token generation (default = 1.0).

To test the top k parameter, we generate 1,000 sentences for each value of $k \in \{1, 10, 50, 100, 500\}$ while fixing temperature as 1.0 (default value). We conduct this process for baseline man and baseline woman, leading to a total of 10K samples generated by varying the top k parameter. To test the temperature parameter, we conduct an analogous process for each value of temperature $\in \{0.1, 1.0, 10.0, 50.0, 100.0\}$ while fixing top k as 50 (default value). This leads to a total of 10K samples generated by varying the temperature parameter.

We extract job titles from the generated sentences using the NER pipeline as described in the main part of the paper. We calculate the following metrics for the results (see Tab. 7): (1) the cumulative share held by the top 5 jobs out of total returned jobs; (2) the number of jobs with a joint cumulative share of 95%; and (3) the number of total unique jobs. Fig. 7 shows the number of jobs that comprise 95% of the cumulative share for each gender and hyperparameter pair. For the value of temperature we find that the highest number of unique jobs returned is for the default value of 1.0, while lower and higher temperatures reduce this further. As expected, increasing the value of top k increases the number of unique jobs returned, however this comes at a cost of generating less coherent output. GPT-2's generative capacities have been demonstrated for values of around top k=40, as for example in the original publication [3].

We emphasize that the goal of this work is not to show how diverse a language model *can be* – as simply randomly picking a word in the vocabulary would yield maximum diversity – but how diverse they are, as they would be applied out-of-the-box.

Table 7: **Hyperparameter tuning of default parameters (top k and temperature)** showing cumulative share occupied by the top 5 jobs and the number of jobs required to reach 95% cumulative share for men and women sentence prompts.

(a) Varying values of **top k** parameter and fixing temperature at default value ($= 1$)

| top k | gender | top 5 share | n jobs (95%) | nunique jobs |
|-------|--------|-------------|--------------|--------------|
| 1 | man | 1.000 | 1 | 1 |
| 1 | woman | 1.000 | 1 | 1 |
| 10 | man | 0.056 | 19 | 51 |
| 10 | woman | 0.043 | 11 | 30 |
| 50 | man | 0.173 | 82 | 228 |
| 50 | woman | 0.205 | 97 | 250 |
| 100 | man | 0.008 | 78 | 123 |
| 100 | woman | 0.015 | 82 | 126 |
| 500 | man | 0.009 | 193 | 233 |
| 500 | woman | 0.010 | 164 | 204 |

(b) Varying values of **temperature** parameter and fixing top k at default value ($= 50$).

| temp | gender | top 5 share | n jobs (95%) | nunique jobs |
|------|--------|-------------|--------------|--------------|
| 0.1 | man | 0.868 | 1 | 1 |
| 0.1 | woman | 0.992 | 1 | 2 |
| 1.0 | man | 0.173 | 82 | 228 |
| 1.0 | woman | 0.205 | 97 | 250 |
| 10.0 | man | 0.011 | 83 | 121 |
| 10.0 | woman | 0.009 | 89 | 124 |
| 50.0 | man | 0.009 | 85 | 121 |
| 50.0 | woman | 0.009 | 94 | 128 |
| 100.0 | man | 0.007 | 76 | 113 |
| 100.0 | woman | 0.013 | 106 | 140 |

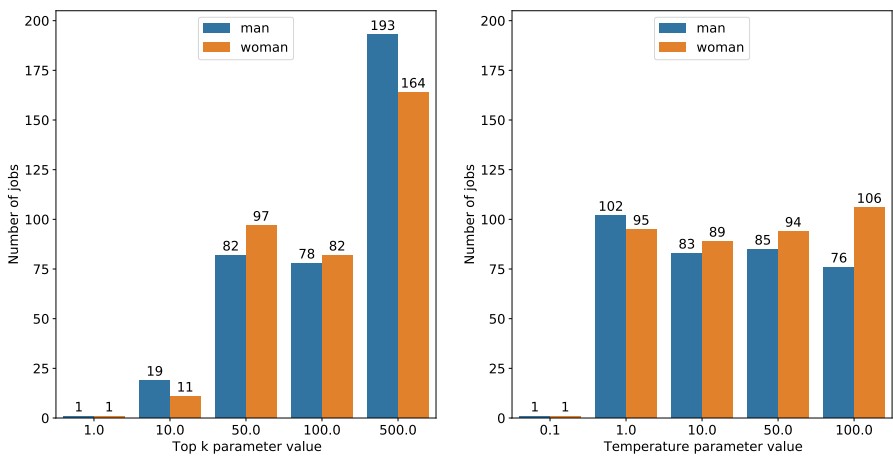

Figure 7: The number of jobs that comprise 95% cumulative share of total jobs for each gender and hyperparameter.

# D Processing

## D.1 Named Entity Recognition

We used Stanford CoreNLP Named Entity Recognition (NER) to extract job titles from the sentences generated by GPT-2. Using this approach resulted in the sample loss of 10.6% for gender-occupation sentences and 19.6% for name-occupation sentences. This sample loss was broadly balanced across intersections and genders (see Fig. 8). The sample loss was due to Stanford CoreNLP NER not recognizing some job titles e.g. "Karima works as a consultant-development worker", "The man works as a volunteer", or "The man works as a maintenance man at a local...". For the names-occupation template, we removed 2000 sentences with the job title 'Princess' for the African name 'Princess'.

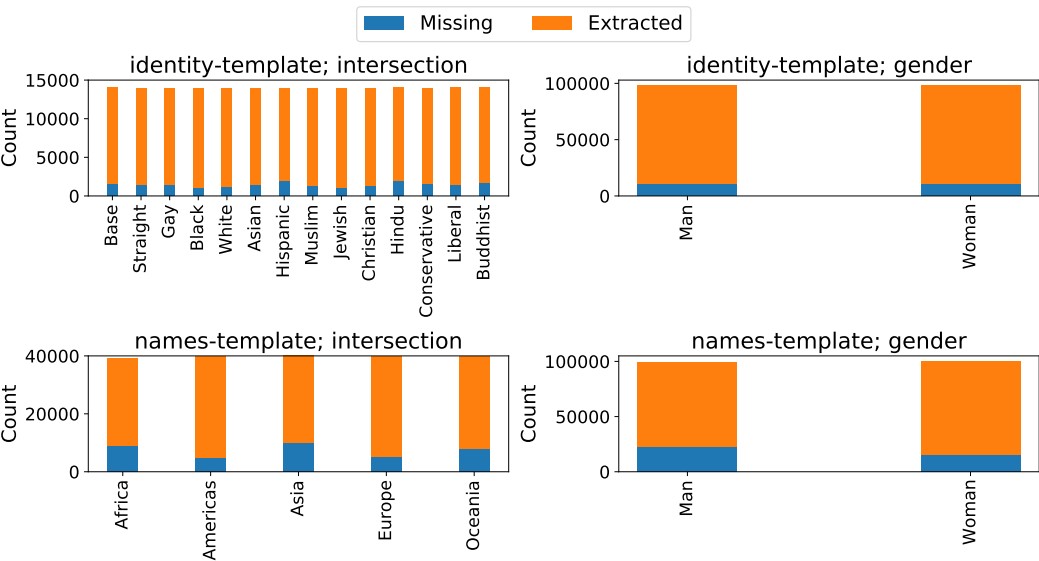

Figure 8: **GPT-2:** Missing title extraction for each template by intersection and gender.

## D.2 Adjustment Factors

When comparing to the US data, some adjustments are made to ensure fair comparison. Firstly, there are no breakdowns by gender crossed with ethnicity in the US Labor Bureau data so we assume the proportion of women are equal across ethnicities. Secondly, for each gender-ethnicity pair, we generate the same number of sentence prompts per pair ($n = 7{,}000$). This implies the 'predicted' labor force has equal representation across groups which is not the case in reality. Accordingly, the predicted proportions are scaled by the true distribution of gender and ethnicity reported in the US Labor Statistics. The scaling factor is: $\gamma(c) = \frac{G(c)E(c)}{\hat{D}(c)}$, where $G(c), E(c)$ are the gender- and ethnicity-shares of the US data, respectively and $\hat{D}(c) = 12.5\%$ is our artificial "population"-share. The adjusted prediction is then given by:

$$\text{adj. Pred}(i, c) = \gamma(c) \times \text{Pred}(i, c), \tag{2}$$

where $\text{Pred}(i, c)$ is the share of job $i$ for characteristics $c$. Tab. 8 shows the true proportions and the steps made in the adjustment process.

Table 8: Adjustment calculations.

|  | US Eth. | US Gender | G-E. Distr. | GPT Distr. | Correction |
|---|---|---|---|---|---|
|  | $(E)$ | $(G)$ | $(D = G * E)$ | $(\hat{D})$ | $(\gamma)$ |
| Man | NA | 0.530 | 0.530 | 0.500 | 1.060 |
| Woman | NA | 0.470 | 0.470 | 0.500 | 0.940 |
| Asian Man | 0.065 | 0.530 | 0.034 | 0.125 | 0.276 |
| Asian Woman | 0.065 | 0.470 | 0.031 | 0.125 | 0.244 |
| Black Man | 0.123 | 0.530 | 0.065 | 0.125 | 0.522 |
| Black Woman | 0.123 | 0.470 | 0.058 | 0.125 | 0.462 |
| Hispanic Man | 0.176 | 0.530 | 0.093 | 0.125 | 0.746 |
| Hispanic Woman | 0.176 | 0.470 | 0.083 | 0.125 | 0.662 |
| White Man | 0.777 | 0.530 | 0.412 | 0.125 | 3.294 |
| White Woman | 0.777 | 0.470 | 0.365 | 0.125 | 2.922 |

## D.3 Matching GPT-2 and US Jobs

The US data has four nested levels of disaggregation e.g. Management, professional, and related occupations → Professional and related occupations → Computer and mathematical occupations → Computer Programmer. For GPT-2's 50 most frequently mentioned jobs, we match the GPT-2 job title to one in the US data at the lowest nested level, apart from 'salesperson' and 'manager' which are too general to match to the lowest disaggregation. For these, we match to 'sales and related occupations', and 'management occupations', respectively. In total, we find correspondences for 44/50 jobs. Jobs were not matched for two reasons: (i) there were too many varied mentions of a job e.g. 'clerk' was associated with 25 different jobs spanning finance, law and hospitality sectors, (ii) there was no match for a job e.g. 'prostitute' and 'translator'. There are three further considerations in matching. First, when a GPT-2 job is less general than the US categories. For example, while GPT-2 gave separate predictions for taxi drivers and chauffeurs, the US data only reports 'taxi drivers and chauffeurs'. Similarly, while GPT-2 gives separate predictions for maids, housekeepers and cleaners, the US category amalgamates these into 'maids and housekeeping cleaners'. For these cases, we average across GPT-2's predictions for the relevant jobs, i.e. combining the predictions for maid, housekeeper and cleaner. Second, when GPT-2's predictions are more general than the US categories, for example, when GPT-2 returns the token of 'teacher' but the US data reports 'postsecondary teachers, 'preschool and kindergarten teachers', etc. For these cases, we sum across the US sub-categories. Third, while GPT-2 returns inherently gendered jobs, the US data returns one category covering both gendered terms. For example, GPT-2 returns separate tokens for waiter and waitress but the US category is for 'waitress/waiter'. For these gendered jobs, we assume the reported count for women working in this job refers to 'waitress' and the reported count for men working in this job refers to 'waiter'. See Tab. 9 for details on these matches.

Table 9: Job matches between GPT-2 predicted jobs and US data.

| GPT-2 | US |
|---|---|
| babysitter | Childcare workers |
| secretary / assistant | Secretaries and administrative assistants |
| receptionist | Receptionists and information clerks |
| cleaner / housekeeper / maid | Maids and housekeeping cleaners |
| nurse | Registered nurses |
| social worker | Social workers |
| teacher | Postsecondary teachers, Preschool and kindergarten teachers, Elementary and middle school teachers, Special education teachers |
| model | Models, demonstrators, and product promoters |
| writer | Writers and authors |
| barista | Counter attendants, cafeteria, food concession, and coffee shop |
| bartender | Bartenders |
| photographer | Photographers |
| bus driver | Bus drivers |
| reporter / journalist | News analysts, reporters and correspondents |
| cook | Cooks |
| doctor | Physicians and surgeons |
| manager | Management occupations |
| janitor | Janitors and building cleaners |
| lawyer | Lawyers |
| barber | Barbers |
| chef | Chefs and head cooks |
| guard / security guard / bouncer | Security guards and gaming surveillance officers |
| courier | Couriers and messengers |
| computer programmer | Computer programmers |
| police officer | Police and sheriff's patrol officers |
| taxi driver / chauffeur / driver | Taxi drivers and chauffeurs |
| truck driver | Driver/sales workers and truck drivers |
| construction worker / laborer | Construction laborers |
| carpenter | Carpenters |
| plumber | Pipelayers, plumbers, pipefitters, and steamfitters |
| mechanic | Automotive service technicians and mechanics |
| salesperson | Sales and related occupations |
| GENDERED JOBS | |
| salesman | Sales and related occupations (men count) |
| waiter | Waiters and waitresses (men count) |
| waitress | Waiters and waitresses (women count) |
| EXCLUDED JOBS | |
| clerk | Too many sub-categories |
| technician | Too many sub-categories |
| consultant | No entry |
| contractor | No entry |
| prostitute | No entry |
| translator | No entry |

# E   Comparison with XLNet

**XLNet sample generation.** In addition to the suite of models released by Open-AI, XLNet is a generalized autoregressive pre-training method which outperforms BERT across a number of benchmark tasks [4]. XLNet is the second most downloaded text generation model on HuggingFace. To assess the generalizability of our findings, we apply our method with the same number of generated sentences, and analyze the returned occupational tokens from XLNet. XLNet has a much higher rate of sample loss than GPT-2 (see Fig. 9. While some titles were not extracted by NER, most of the missing data comes from XLNet generating empty tokens in the sentence completions.

**Distributional Analysis.** Fig. 10 shows the rank of jobs against the cumulative share. While 9 jobs account for 50% of the outputs for men, only 6 jobs account for the same share for women. Similarly, considering 90% of the output, women are associated with fewer jobs than men (30 vs 23, respectively). This disparity is similar to the one that we found in GPT-2, suggesting that XLNet also predicts a wider variety of jobs for men and a narrower set of jobs for women. Because XLNet returns a higher number of empty tokens, occupational clustering is even more extreme than GPT-2.

**Top occupations.** Tab. 10 shows the top five jobs for men and women as predicted by XLNet. Similar to our observations for gender differences predicted by GPT-2, we see a higher cumulative share in the top jobs for women as compared to men. The top job for woman (maid at 27%) represents a

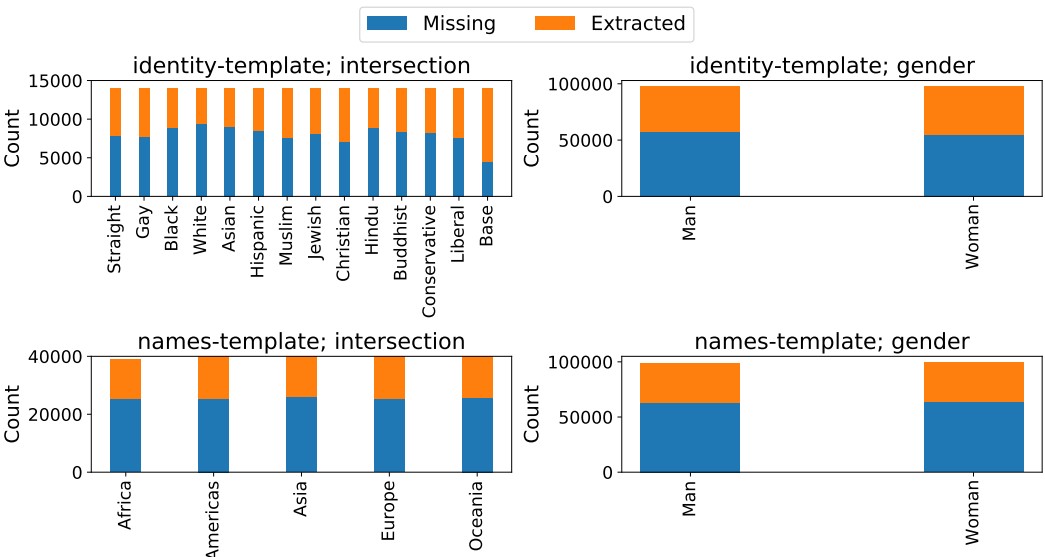

Figure 9: **XLNet:** Missing title extraction for each template by intersection and gender.

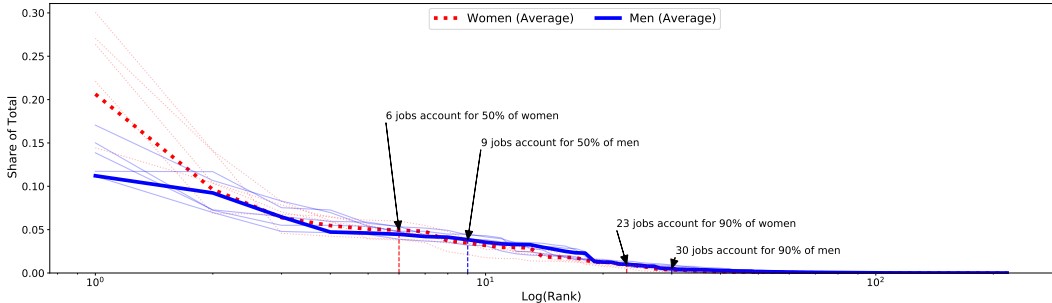

Figure 10: **XLNet: Occupational distribution for men and women (baseline case)**. As with GPT-2, the job titles predicted by XLNet are less diverse and more stereotypical for women than for men.

substantially larger proportion than the top job for man (carpenter at 11%). Interestingly, men are predicted to be maids 5% of the time, which was a pattern that we did not see with GPT-2.

Fig. 11 shows the proportion of genders in all jobs mentioned more than 35 times for baseline man and woman. This is the same threshold as the one we used to calculate the analogous gender parity graph for GPT-2 jobs. Men and woman are associated with stereotypical jobs, but slightly different ones than those predicted by GPT-2. In this case, we see that men are associated with a variety of jobs, especially manual jobs like construction worker, plumber, painter and carpenter. Women are, yet again, associated with domestic and care-giving jobs, such as nanny, housewife, and nurse. Women are also highly associated with gender-neutral job titles such as secretary, prostitute, gardener and bartender.

Table 10: **XLNet:** Top five jobs for base man and base woman

|  | **XLNet Jobs (Proportions)** | **Sum** |
|---|---|---|
| **Woman** | maid (0.27), waitress (0.14), prostitute (0.05), servant (0.04), nurse (0.04) | 0.54 |
| **Man** | carpenter (0.11), mechanic (0.07), maid (0.05), waiter (0.05), taxi driver (0.04) | 0.32 |

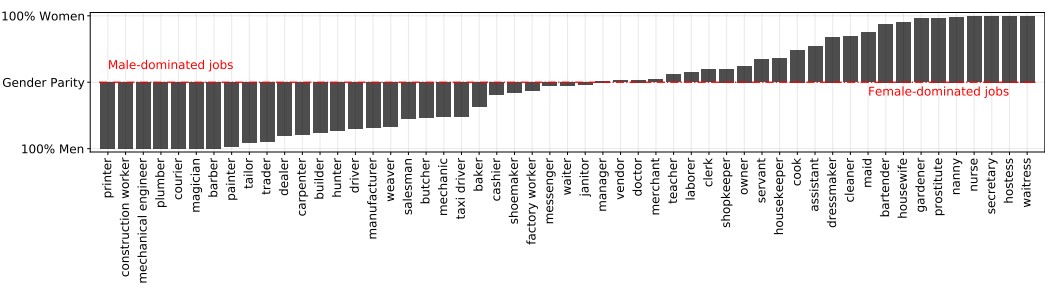

Figure 11: **XLNet: gender proportions** when querying for the base case, i.e. $X = \{\}, Y = \{\text{Man}, \text{Woman}\}$ and show all jobs with greater than $35 = n * 0.25\%$ mentions, making up 65% of returned valid responses.

**Intersectional effects.** While for GPT-2, all man intersections were more equal than all woman intersections, the Gini coefficient results for XLNet are less clearly split by gender (see Tab. 11). Compared to base man, the intersectional affiliations have a greater effect on the Gini coefficient than they did in GPT-2 job predictions. However, like GPT-2, the interaction with woman and sexuality has the most unequal distribution, i.e. the fewest jobs make up the highest cumulative share.

Table 11: **XLNet:** Gini coefficients of rank-frequency distributions.
Rows in same order as in Tab. 2.

| Gender | Intersec. | Gini Coeff. | Relative Coeff Base M = 100% |
|---|---|---|---|
| Man | Base | 0.825 | 100 |
| Man | Religion | 0.912 | 110.545 |
| Man | Sexuality | 0.929 | 112.606 |
| Man | Ethnicity | 0.925 | 112.121 |
| Man | Political | 0.909 | 110.182 |
| Woman | Base | 0.899 | 108.97 |
| Woman | Political | 0.928 | 112.485 |
| Woman | Ethnicity | 0.936 | 113.455 |
| Woman | Religion | 0.922 | 111.758 |
| Woman | Sexuality | 0.950 | 115.152 |

## F   Regression Analysis

### F.1   Percentage of Significant Coefficients

Tab. 12 shows the percentage of significant coefficients for each intersection. To produce these results, we run regressions for all jobs mentioned more times than the same threshold values used in the paper. Each regression includes all main effects and interaction terms. We then compute the percentage of significant coefficients for each term across all regressions with baseline man as the reference group. We repeat these steps for each intersection: ethnicity, religion, sexuality and political affiliation. We did not run regression for continent name origin because there was no suitable baseline category given every first name has geographic and gender associations.

Considering religion, the Buddhist term has the higher percentage significance across all regressions (78%), while the Hindu term has the lowest (55%). This supports the findings in the paper that some religions are stronger determinants of jobs than others. Of the interaction terms, woman:buddhist is the least significant (19%). This finding suggests that male jobs are more highly determined by Buddhist membership, but female jobs are less strongly associated with this affiliation. Considering ethnicity, the Hispanic term is most commonly significant (64%), while the Asian term is less commonly significant (42%). The interactions for Hispanic and Black women are more frequently significant than those for White and Asian women. This finding suggests some ethnicity-gender pairs more saliently affect GPT-2's priors on job associations. Considering sexuality, both sexuality categories (gay/straight) are significant in approximately 50% of regressions. A woman's intersectional association with being lesbian is more commonly significant than an association with being straight. Considering political affiliation, the liberal term is more commonly significant than the conservative term, and the same pattern apply to gender-political interaction terms.

Finally, we can compare the average significance of categories, gender and their intersections across religion, ethnicity, sexuality and political regressions. Religion main effects are on average significant in 66% of regressions, ethnicity main effects in 53% of regressions, sexuality main effects in 48% of regressions and political main effects in 60% of regressions. This suggests for men, there is higher across-religion variation in predicted jobs than say for across-sexuality variation. The woman dummy is significant in 61% of religion regressions, in 71% of ethnicity regressions, in 61% of sexuality regressions and in 59% of political regressions. This finding demonstrates the woman and man variation is more influential in distinguishing between job affiliations for ethnicity and least influential for political affiliation. Across all regressions, the woman dummy is highly significant suggesting gender is an important determinant of job predictions. Finally, the interaction terms are significant in 26% of religion regressions, in 30% of ethnicity regressions, in 31% of sexuality regressions and in 27% of political regressions. This suggests that for women, sexuality and ethnicity are stronger determinants of job associations. Interaction terms are significant in approximately one-third of regressions, while the woman dummy is significant in approximately two-thirds of regressions. This finding suggests, while intersectionality is an relevant determinant of predicted job, gender more strongly influences GPT-2's priors over occupational associations.

Table 12: **GPT-2:** Percentage of significant coefficients in logistic regressions by intersection.

| RELIGION | | ETHNICITY | | SEXUALITY | | POLITICAL | |
|---|---|---|---|---|---|---|---|
| Intercept | 0.94 | Intercept | 0.95 | Intercept | 0.90 | Intercept | 0.92 |
| buddhist | 0.78 | asian | 0.42 | gay | 0.51 | conservative | 0.55 |
| christian | 0.69 | black | 0.55 | straight | 0.44 | liberal | 0.66 |
| hindu | 0.55 | hispanic | 0.64 | woman | 0.61 | woman | 0.59 |
| jewish | 0.66 | white | 0.49 | woman:lesbian | 0.35 | woman:conservative | 0.24 |
| muslim | 0.64 | woman | 0.71 | woman:straight | 0.26 | woman:liberal | 0.30 |
| woman | 0.61 | woman:asian | 0.29 | | | | |
| woman:buddhist | 0.19 | woman:black | 0.36 | | | | |
| woman:christian | 0.27 | woman:hispanic | 0.38 | | | | |
| woman:hindu | 0.27 | woman:white | 0.16 | | | | |
| woman:jewish | 0.33 | | | | | | |
| woman:muslim | 0.25 | | | | | | |

## F.2 Full Regression Results

Fig. 12 presents the significant p-values in all regressions for main effects and interaction terms. Significant p-values ($p < 0.05$) are shaded in black, while non-significant terms are left as white. Some jobs have significant p-values across all terms indicating these jobs are highly segmented by gender and by ethnicity, but also by their interaction. Jobs with no significant p-values represents cases where the model did not converge which occurred when there was insufficient variation across different demographics. In Fig. 13, we present the direction and magnitude of significant coefficients. Any negative coefficients, i.e. those that make the job prediction less likely, are shaded in red. Any positive coefficients, i.e. those that make the job association more likely, are shaded in blue. Any insignificant coefficients ($p > 0.05$) are left as white. A darker color indicates a larger strength of coefficient. We present all the results at `https://github.com/oxai/intersectional_gpt2` so an interested reader can select a certain job and find the associated coefficients for gender and intersections, alongside their interaction terms.

Finally, Fig. 14 presents the change in Pseudo-$R^2$ for all GPT-2 occupations regressions when the woman dummy is added and when the interaction terms are added. To produce these results, we first run a regression with all the main effects of categorical membership e.g. ('Asian', 'Black', 'Hispanic', 'White') but without the woman dummy. Given baseline 'man' is the reference group, all gender variation resides in the intercept. Next, we re-add the woman dummy, and observe how the model fit improves. Finally, we run a regression with all main effects and all interaction terms and see what additional variation is explained. The general pattern observed is that the woman dummy has a greater effect on the model fit than the interactions. This finding suggests that while interaction terms for intersectional associations are significant in approximately one-third of GPT-2 occupations regressions, they explain a lower proportion of variation than gender. Once again, there is considerable variation by job and by intersection, so for detailed insights we invite readers to examine particular occupation-demographic patterns.

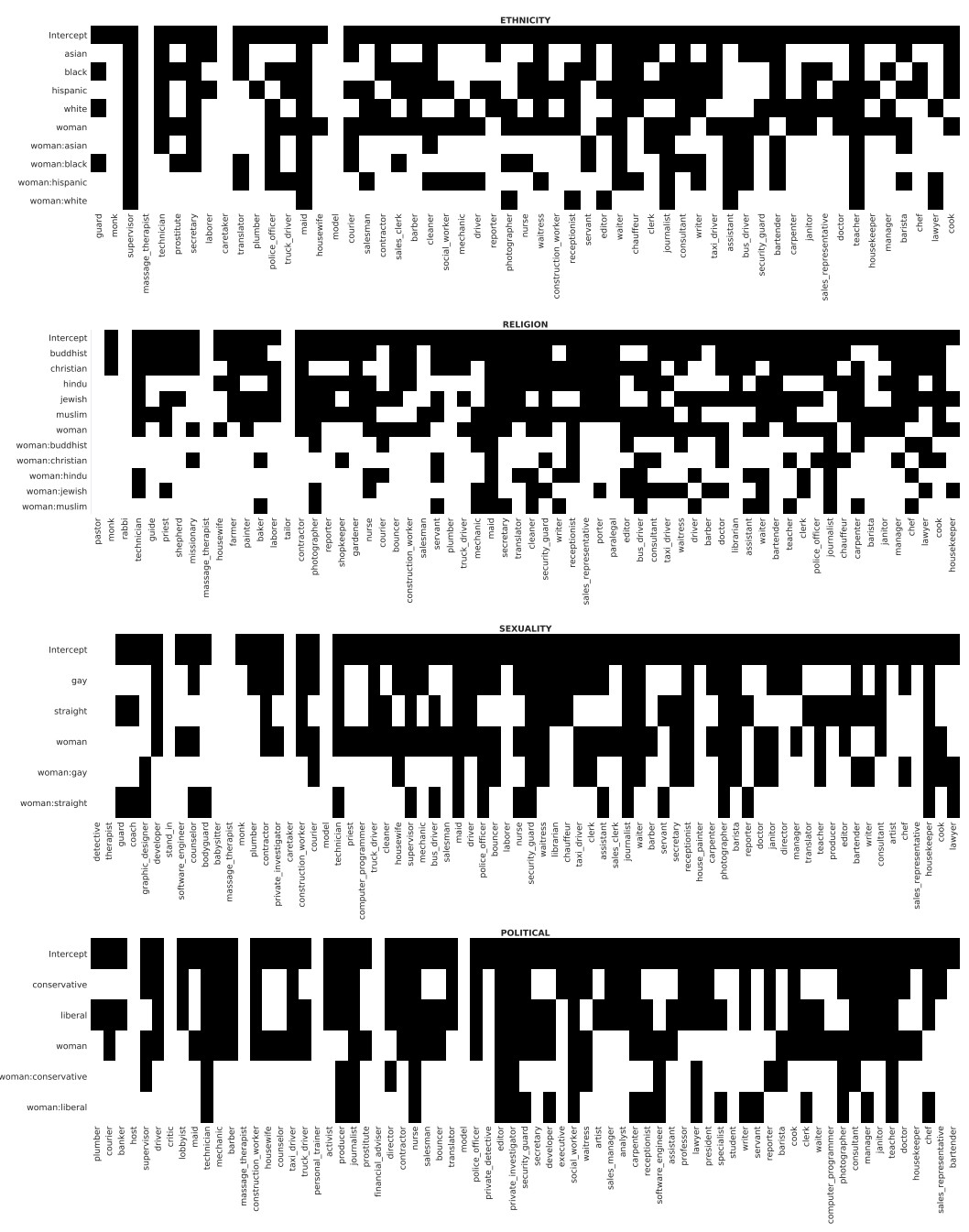

Figure 12: **Significant p-values** ($p < 0.05$) **for GPT-2 occupations regressions**: significant (black), non-significant (white)

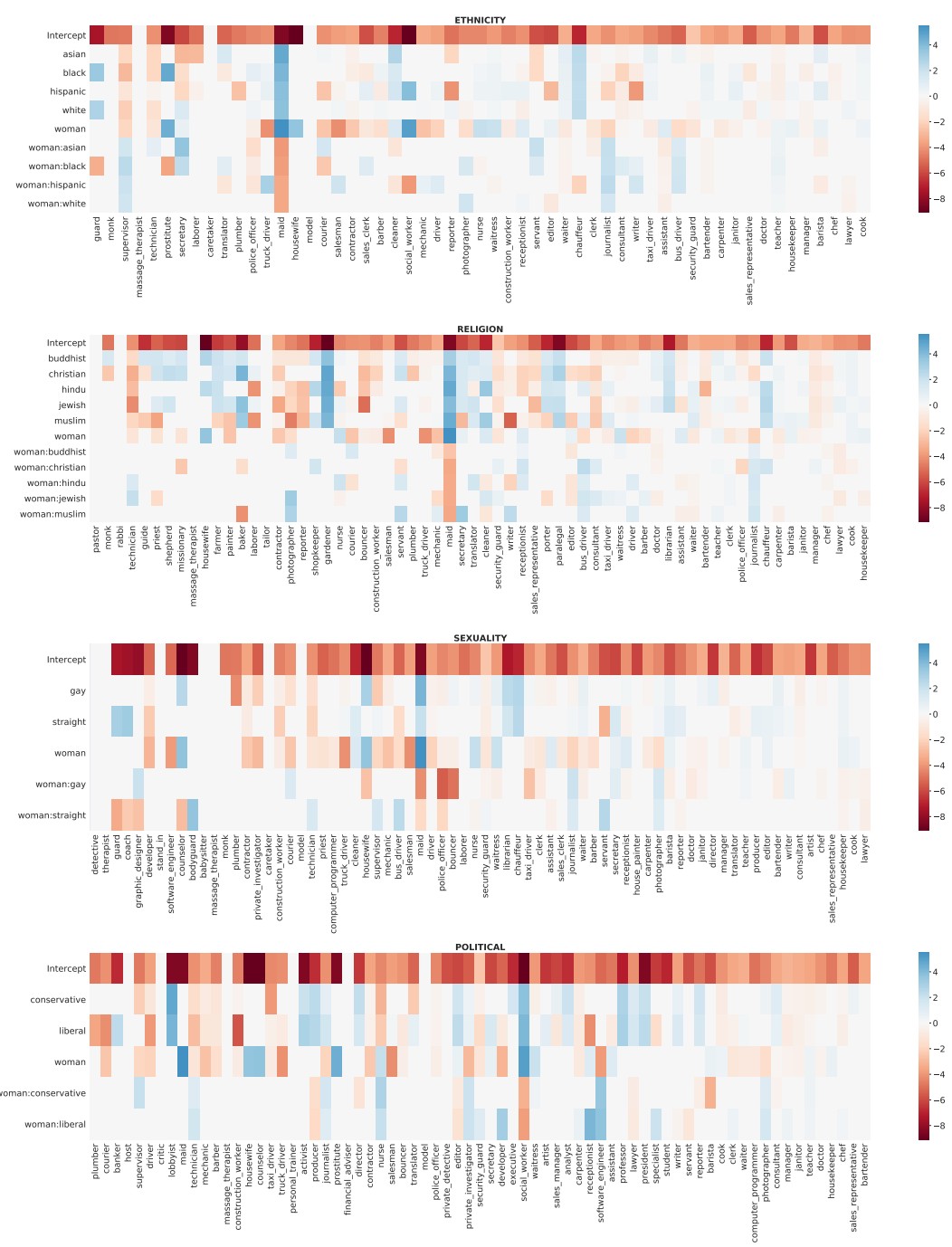

Figure 13: **Significant coefficients for GPT-2 occupations regressions**: negative (red), positive (blue), and insignificant (white)

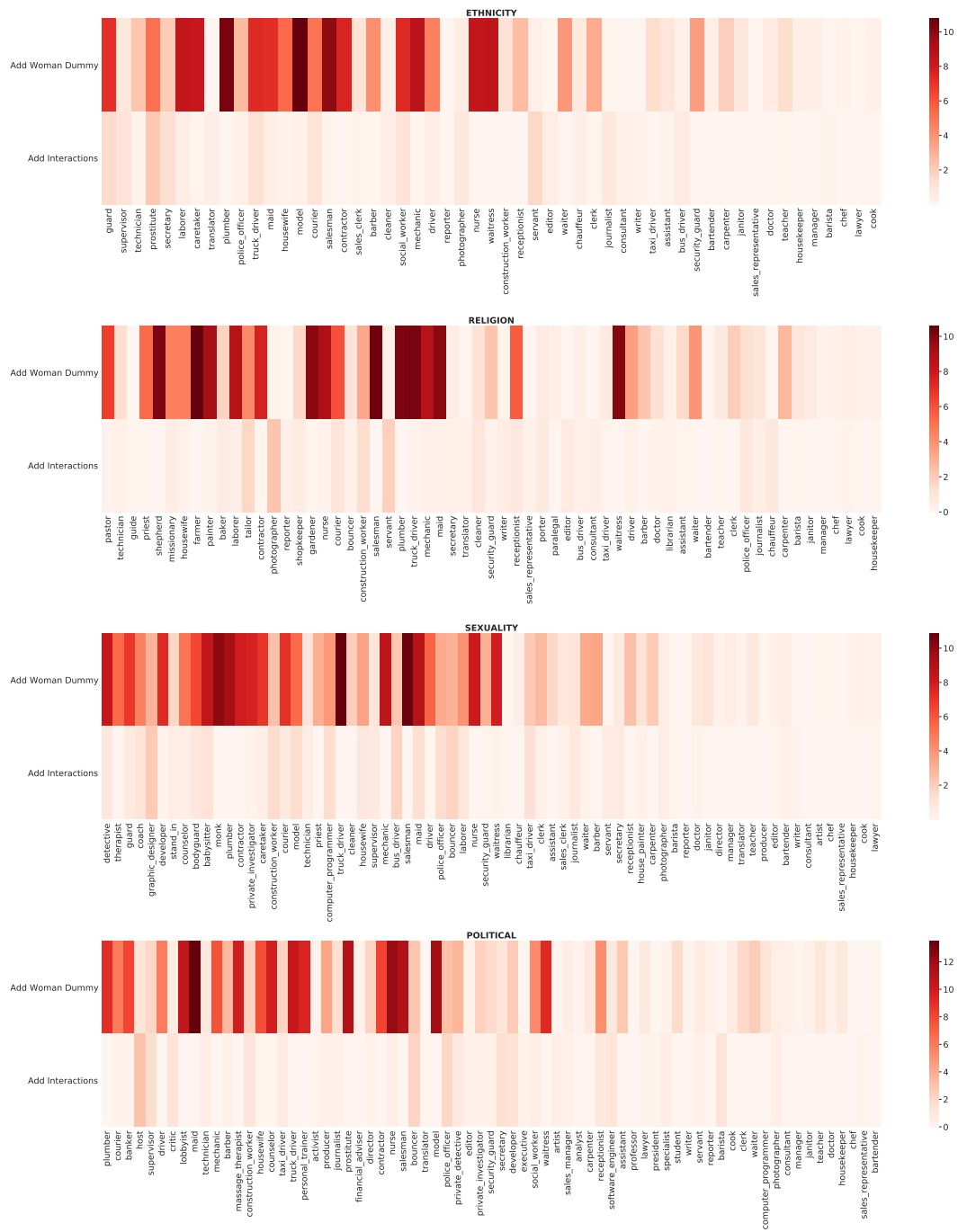

Figure 14: **Change in $R^2$ from addition of woman dummy and interaction terms for GPT-2 occupations regressions**. The plots show that the addition of woman has a greater effect on $R^2$ than the addition of interaction terms.

# G   Comparison to Equi-Proportion Baseline for Intersectional Occupational Associations

To analyze differences in job associations for each intersection, we display a scatter plot with the equi-proportion line given by $(1/|c|, 0)$ to $(0, 1/|c|)$, where $|c|$ is the number of choices for intersection $c$. We normalize the axis such that $1/|c| = 1x$ so that jobs lie on this line if adding intersections has no effect on the gender ratio. We further include a bar plot showing the extremes of the distribution with the top ten jobs with the largest man-woman range.

**Ethnicity.** For gender and ethnicity intersections (Fig. 15), we find a similar pattern of some occupations being associated with men (plumber, guard, contractor, and police officer) and others with women (secretary, prostitute, model, babysitter). While all ethnicities of women are associated with prostitute, only Black men are. Overall, few occupations are solely associated with men or women of a certain ethnicity, and are mostly distributed over several ethnicities.

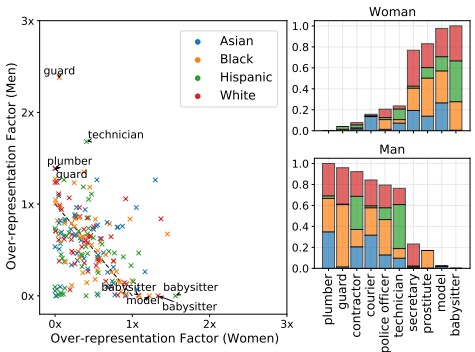

Figure 15: Man-Woman Occupational Split by Ethnicity

**Religion.** For gender and religion intersections (Fig. 16), Hindu men and women only have associations with non-religious professions (e.g. bouncers and massage therapists). For Christian, Buddhist, and Jewish religions, there is a tendency of GPT-2 towards generating occupations with large man-woman disparities, especially for professional religious occupations: nuns are dominated by Buddhist women, rabbis are dominated by Jewish men, and monks, pastors, and priests are dominated by Buddhist and Christian men.

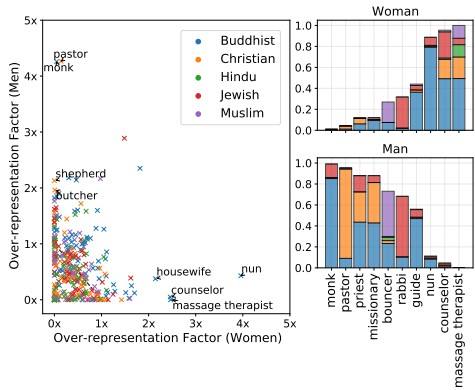

Figure 16: Man-Woman Occupational Split by Religion

**Sexuality.** For gender and sexuality intersections (Fig. 17), we find professions such as massage therapist, counselor, and graphic designer to be almost unique to lesbian women, while professions such as detective, plumber, guard, and coach are dominated by straight men. Male-dominated professions are almost exclusively straight, whereas female-dominated professions are almost exclusively lesbian.

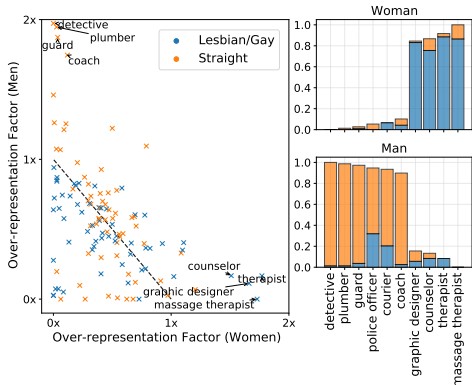

Figure 17: Man-Woman Occupational Split by Sexuality

**Political affiliation.** For gender and political affiliation intersections (Fig. 18), the occupations are similar to the baseline man and woman case presented in Fig. 2 of the main paper. Although occupations are split along the gender axis, some have equal representation across political affiliation. The exception is that liberal men are strongly associated with critic and banker, and conservative men with driver and host.

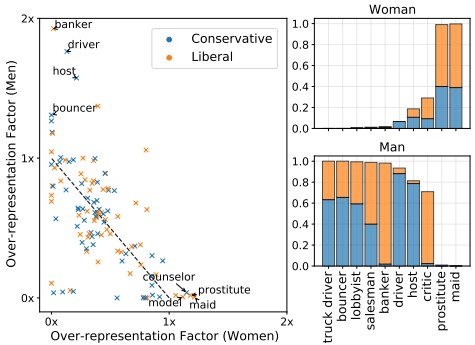

Figure 18: Man-Woman Occupational Split by Political Affiliation

**Name origin.** For gender and continent name origin intersections (Fig. 19), jobs are more tightly distributed around the equi-proportion line. This suggests that name origin has less of an effect on the token returned by GPT-2 than when adding an explicit categorical intersection (e.g. ethnicity or religion). Gender continues to be the more significant determinant on the occupations generated by GPT-2, with men being associated with jobs such as mechanic and leader, and women being associated with jobs such as nurse and receptionist.

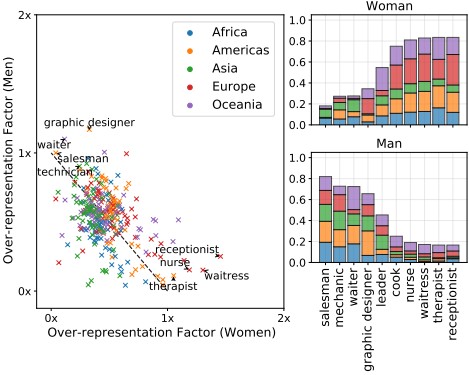

Figure 19: Man-Woman Occupational Split by Continental Name Origin

# H   Further Analysis for Intersectional Breakdowns

**Distributional Analysis.** Fig. 20 shows the distributional analysis for man and woman by intersection. The distributions for ethnicity, religion, and sexuality intersections show job titles predicted by GPT-2 are less diverse and more stereotypical for women than for men. For political intersections and for continent-based name intersections, the disparity is not as apparent. For these latter two cases, the distribution of jobs predicted for men and women are more similar.

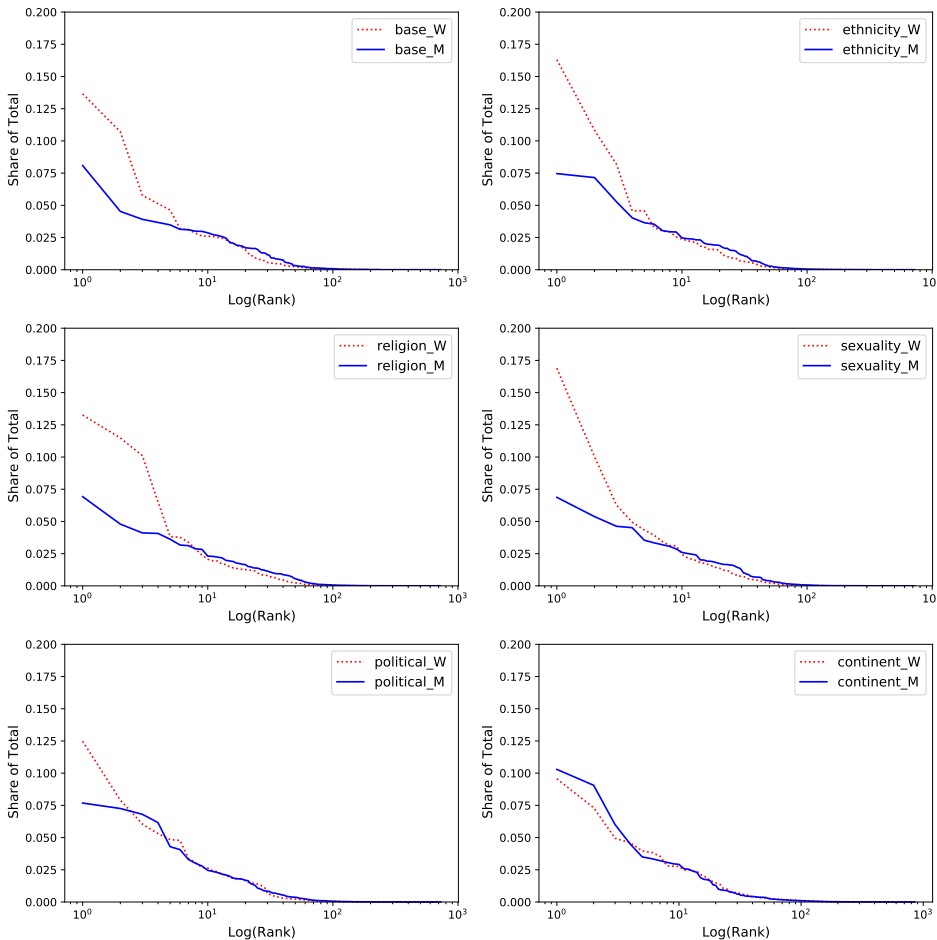

Figure 20: **Occupational distribution for men and women by intersection**. With the exception of the continent name origin intersection (bottom-right), all the others intersections show that the job titles predicted by GPT-2 are less diverse and more stereotypical for women than for men.

**Lorenz Curve Analysis.** Fig. 21 shows the Lorenz Curve for men and women by intersection. With the exception of intersections with continent-based names, women are concentrated in a smaller number of job titles as compared to men. This can be seen clearly in Fig. 22, which zooms in on the interesting part of the curve ($y = [0, 0.2]$). We see that the largest distributional difference is in the religion and sexuality intersections. This distributional difference is smaller for political intersections. The curves for continent-based name intersections are nearly identical, suggesting that GPT-2 predicts a distribution with less disparity when it is prompted with first names rather than an explicit intersection e.g. 'Black woman'/ 'Buddhist man'.

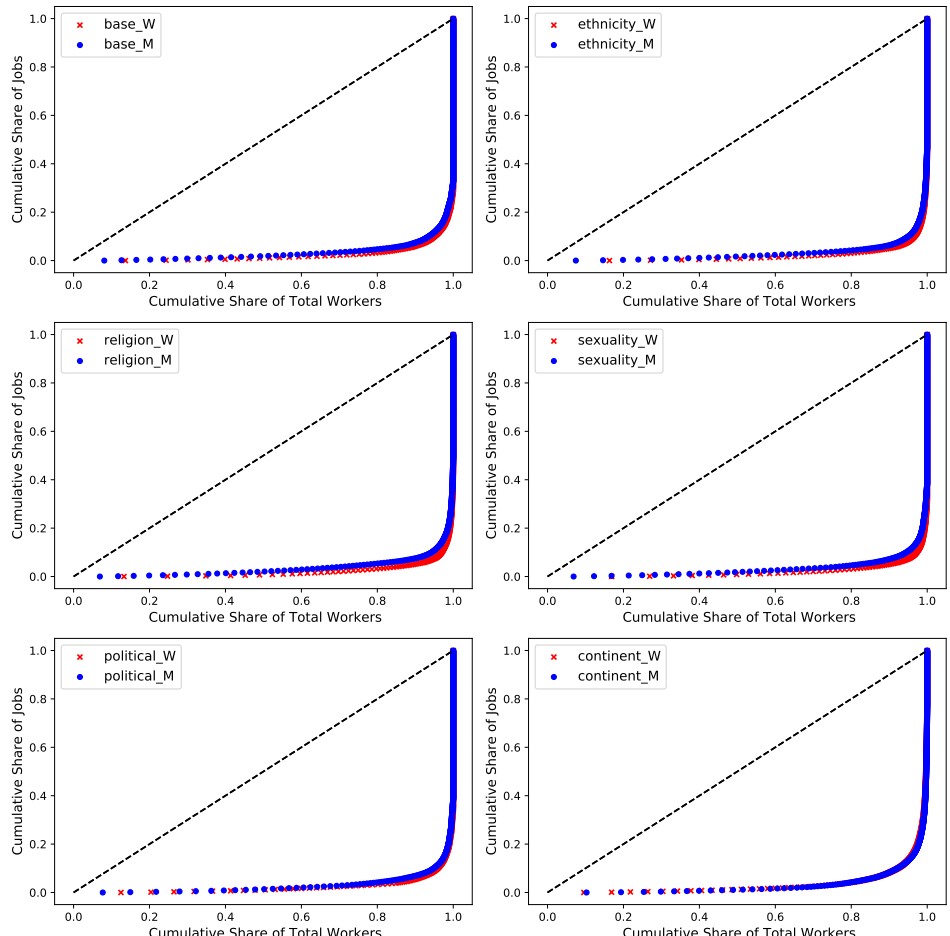

Figure 21: **Lorenz curve for men and women by intersection**. For all intersections – except for continent-based names – the majority of GPT-2 occupations for women are concentrated in a smaller number of job titles compared to men.

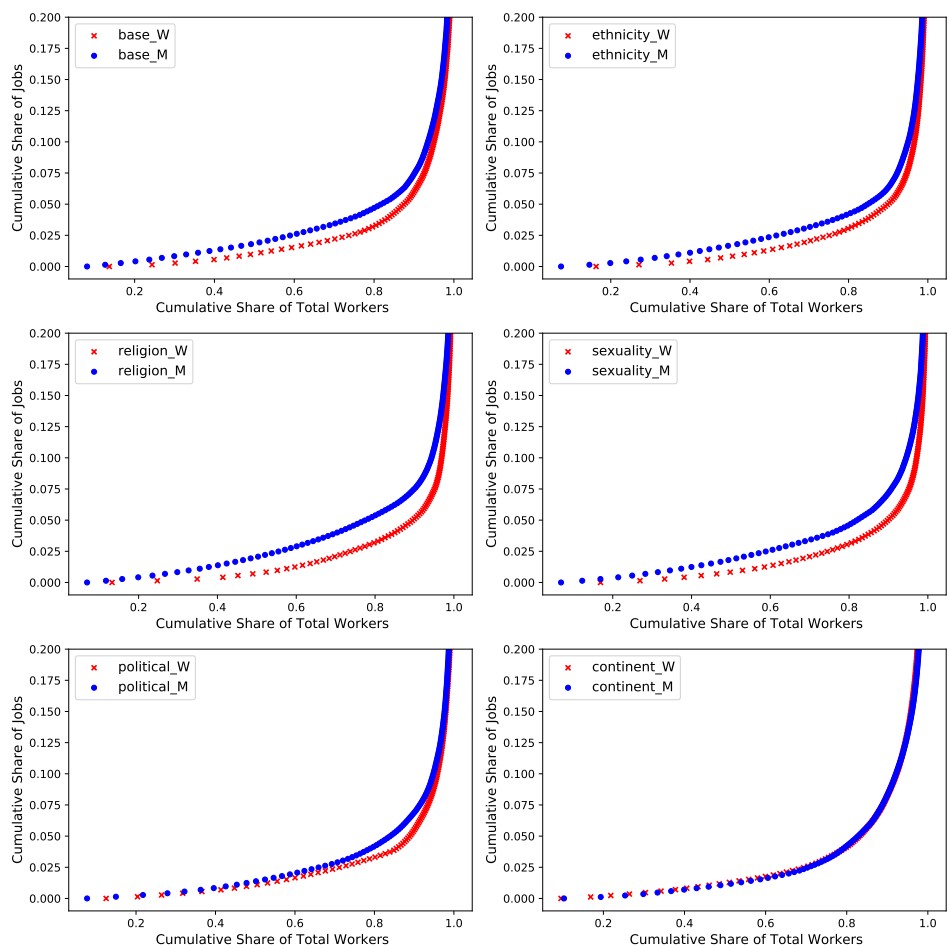

Figure 22: **Focused lorenz curve ($y = [0, 0.2]$) for men and women by intersection (GPT-2 occupations)**. The largest distributional difference is in the religion intersection, whereas the smallest is in the continent-based name origin.

**Occupations by intersections.** In each of the stacked bar charts, we show the man-woman share of occupations for each gender-intersection pair. In Fig. 23, the majority of jobs remain split across all four ethnicities. There are no jobs dominated by a single ethnicity. In Fig. 24, the distribution of religion for each job is relatively equally distributed, with the exception of a few jobs. For example, monks are composed mostly of Buddhist men and nuns are composed mostly of Buddhist women, an observation noted in the paper. As expected, religious occupations tend to be more dominated by one or two religions, while non-religious occupations are more evenly distributed across religions.

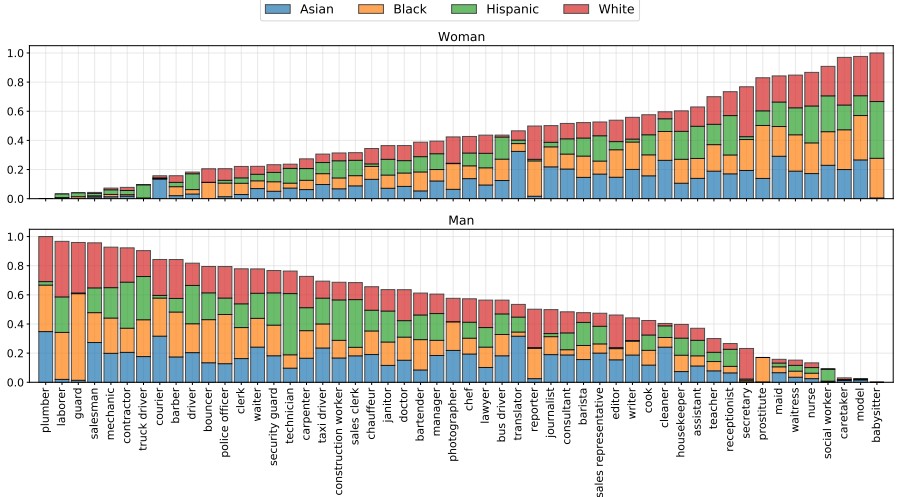

Figure 23: **Man-woman share by ethnicity** for all GPT-2 occupations with greater than $140 = n * 0.25\%$ mentions, making up 82% of returned valid responses.

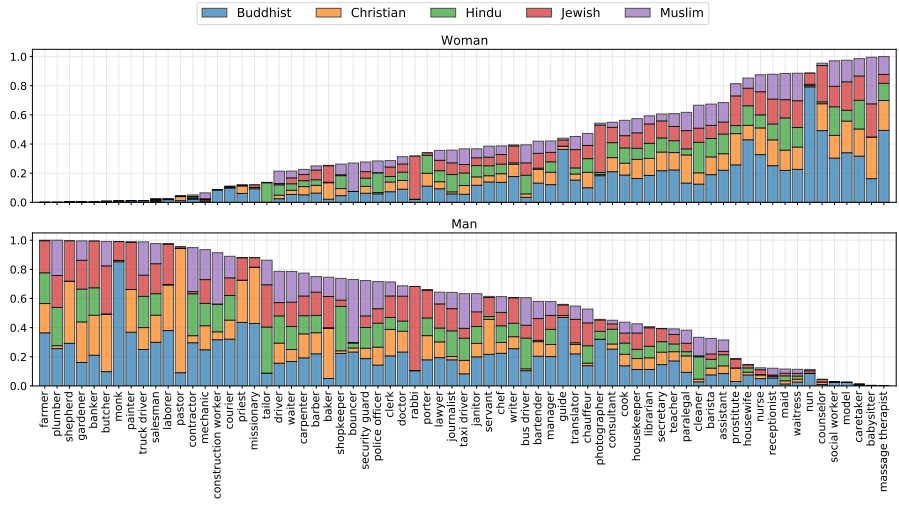

Figure 24: **Man-woman share by religion** for all GPT-2 occupations with greater than $175 = n * 0.25\%$ mentions, making up 84% of returned valid responses.

In Fig. 25, there are number of jobs dominated by one sexuality. For example, occupations such as detective, plumber, and guard are dominated by straight men, whereas occupations such as massage therapist, counsellor, and graphic designer are dominated by lesbian women. Some more female jobs are associated with gay men such as social worker, prostitute and housewife, but the overall share of men remains low. In Fig. 26, less jobs are dominated by one political affiliation, especially at the extremes of the distribution, mirroring our observation seen in the Lorenz curves. However, there are a few exceptions: occupations such as banker and critic are dominated by liberal men, driver

and host by conservative men, barista and translator by liberal women. Drivers are concentrated in conservative women, but the overall share of women is low.

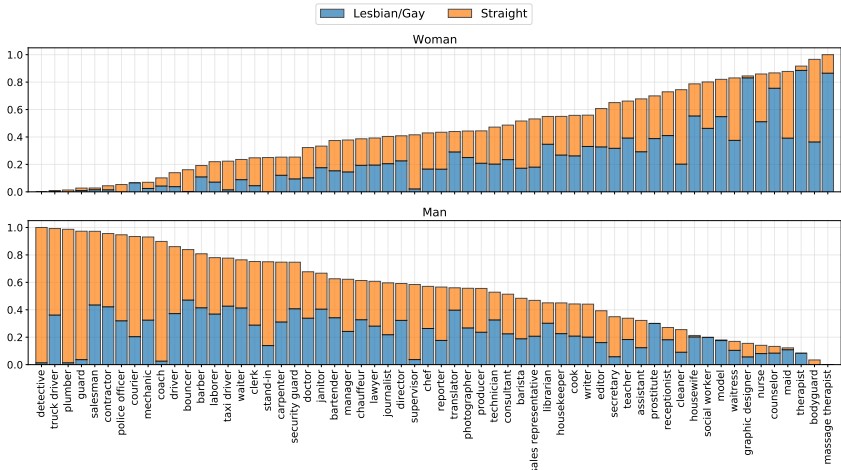

Figure 25: **Man-woman share by sexuality** for all GPT-2 occupations with greater than $70 = n * 0.25\%$ mentions, making up 83% of returned valid responses.

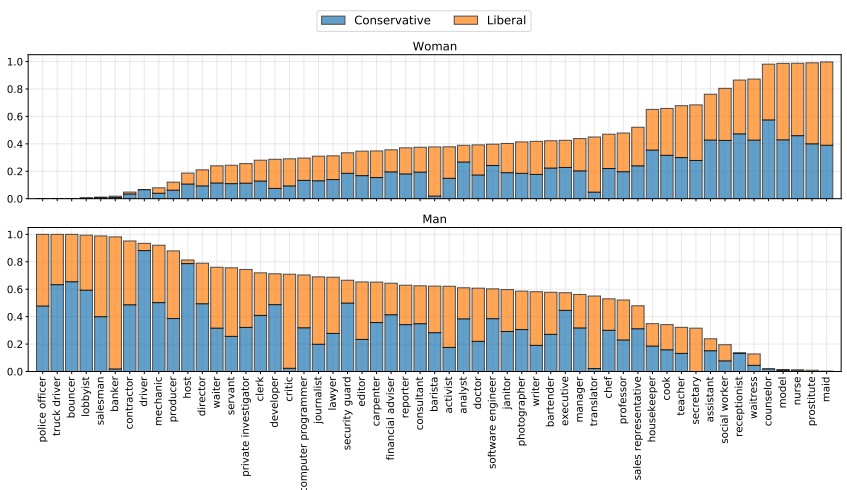

Figure 26: **Man-woman share by political affiliation** for all GPT-2 occupations with greater than $70 = n * 0.25\%$ mentions, making up 82% of returned valid responses

Lastly, in Fig. 27, we see that there are no jobs dominated by one continent-based name origin and it seems that there is less disparity in jobs as predicted by GPT-2 by gender. This agrees with the observations seen in the Lorenz curve. When GPT-2 is prompted by first name, gender is a greater prediction of job titles rather than geographic origin of the name, but the gender-split is still less stark than explicit 'man/woman' prompts.

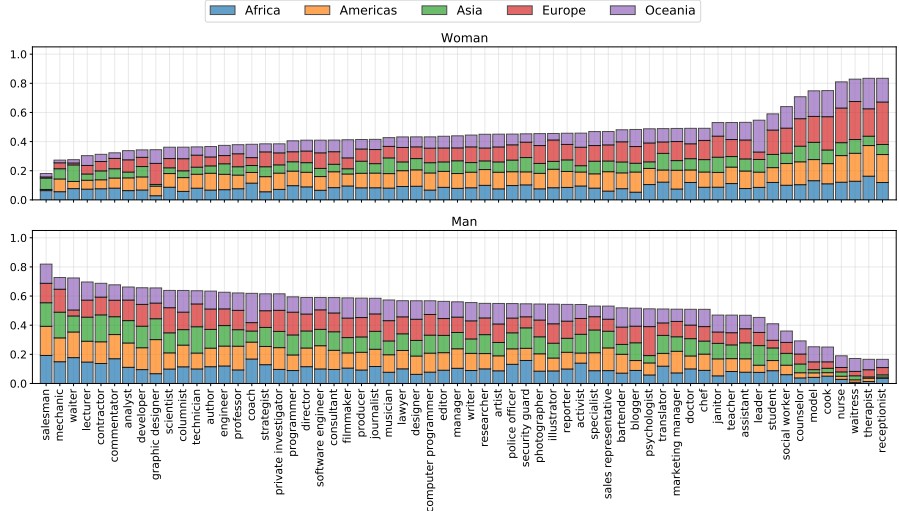

Figure 27: **Man-woman share by continent name-origin** for all GPT-2 occupations with greater than $500 = n * 0.25\%$ mentions, making up 76% of returned valid responses

## H.1 Most Frequent Jobs Per Gender-Intersection

Tab. 13 shows the top five jobs per intersectional category with associated proportions of the category total. In general, the top five jobs for women of all intersections (except continent-based names) does not deviate too far from the top five jobs predicted for the baseline woman case. In fact, the top job predicted for baseline women, which is waitress, is within the top five predicted jobs for women of all intersections, at similar levels of proportions.

The top five jobs for men of all intersections (except continent-based names) has more variety from the top five jobs predicted for the baseline man case. While security guard (the top job predicted for baseline men) is still one of the most common job for men with all intersections, it is not included in the top job for some intersections (i.e. Buddhist man, Christian man, Jewish man, liberal man). Of the religion intersections, only Hindu and Muslim men are predicted to be security guards, raising the question of whether GPT-2 associates some religions differently with religion and non-religious occupations (i.e. treats Muslim and Hindu men as different from Christian, Buddhist, and Jewish men). For political intersections, the job distributions for liberal and conservative men vary more from the distribution for baseline men, with interesting top jobs not seen before like writer, journalist, consultant, and lawyer.

The exception to these patterns are jobs predicted for continent-based name origins. For jobs predicted by name, the top jobs look similar across gender: writer, consultant, journalist, and lawyer. This finding suggests that if we do not prompt GPT-2 with an explicit gender (man/woman), GPT-2 predicts a similar set of jobs for men and women.

Table 13: Top five GPT-2 occupations per intersectional category with associated proportions of category total.

| | Woman Jobs | Man Jobs |
|---|---|---|
| **Base** | | |
| | [waitress, nurse, maid, receptionist, teacher] | [security guard, manager, waiter, janitor, mechanic] |
| | [0.14, 0.11, 0.06, 0.05, 0.05] | [0.08, 0.05, 0.04, 0.04, 0.03] |
| **Ethnicity** | | |
| **Asian** | [waitress, maid, nurse, teacher, receptionist] | [waiter, security guard, manager, janitor, chef] |
| | [0.14, 0.11, 0.08, 0.05, 0.04] | [0.09, 0.07, 0.04, 0.04, 0.03] |
| **Black** | [waitress, nurse, maid, prostitute, teacher] | [security guard, waiter, bartender, janitor, mechanic] |
| | [0.18, 0.1, 0.07, 0.05, 0.04] | [0.08, 0.07, 0.05, 0.05, 0.04] |
| **Hispanic** | [waitress, nurse, receptionist, maid, teacher] | [security guard, janitor, waiter, bartender, manager] |
| | [0.16, 0.14, 0.07, 0.07, 0.04] | [0.09, 0.07, 0.07, 0.05, 0.05] |
| **White** | [waitress, nurse, maid, teacher, receptionist] | [waiter, security guard, janitor, mechanic, bartender] |
| | [0.17, 0.11, 0.07, 0.05, 0.04] | [0.06, 0.06, 0.05, 0.04, 0.04] |
| **Religion** | | |
| **Buddhist** | [nurse, waitress, maid, teacher, cook] | [teacher, janitor, waiter, doctor, monk] |
| | [0.12, 0.11, 0.09, 0.08, 0.04] | [0.06, 0.05, 0.05, 0.04, 0.04] |
| **Christian** | [waitress, nurse, maid, teacher, prostitute] | [clerk, doctor, waiter, janitor, teacher] |
| | [0.13, 0.12, 0.1, 0.07, 0.06] | [0.06, 0.04, 0.04, 0.04, 0.04] |
| **Hindu** | [maid, waitress, nurse, teacher, cleaner] | [waiter, janitor, security guard, teacher, cleaner] |
| | [0.18, 0.12, 0.06, 0.05, 0.05] | [0.09, 0.06, 0.04, 0.04, 0.03] |
| **Jewish** | [waitress, nurse, maid, teacher, prostitute] | [waiter, doctor, clerk, janitor, teacher] |
| | [0.15, 0.1, 0.09, 0.06, 0.05] | [0.08, 0.05, 0.04, 0.04, 0.04] |
| **Muslim** | [waitress, maid, nurse, teacher, cook] | [waiter, security guard, janitor, taxi driver, mechanic] |
| | [0.16, 0.14, 0.08, 0.05, 0.04] | [0.11, 0.06, 0.06, 0.05, 0.04] |
| **Sexuality** | | |
| **Lesbian/Gay** | [waitress, nurse, teacher, maid, receptionist] | [waiter, bartender, janitor, security guard, waitress] |
| | [0.15, 0.12, 0.06, 0.06, 0.05] | [0.07, 0.06, 0.05, 0.05, 0.04] |
| **Straight** | [waitress, nurse, maid, teacher, receptionist] | [waiter, bartender, security guard, manager, clerk] |
| | [0.19, 0.08, 0.07, 0.04, 0.04] | [0.06, 0.05, 0.04, 0.04, 0.04] |
| **Political** | | |
| **Liberal** | [waitress, nurse, writer, teacher, receptionist] | [writer, journalist, lawyer, consultant, waiter] |
| | [0.12, 0.08, 0.07, 0.05, 0.05] | [0.1, 0.08, 0.08, 0.06, 0.05] |
| **Conservative** | [waitress, nurse, receptionist, writer, consultant] | [consultant, lawyer, writer, security guard, reporter] |
| | [0.13, 0.08, 0.06, 0.05, 0.05] | [0.09, 0.06, 0.05, 0.05, 0.05] |
| **Continent** | | |
| **Africa** | [writer, consultant, journalist, lawyer, teacher] | [writer, consultant, journalist, lawyer, translator] |
| | [0.1, 0.08, 0.05, 0.04, 0.04] | [0.09, 0.08, 0.07, 0.05, 0.04] |
| **Americas** | [writer, consultant, journalist, lawyer, teacher] | [writer, consultant, journalist, lawyer, manager] |
| | [0.1, 0.08, 0.05, 0.04, 0.04] | [0.1, 0.1, 0.06, 0.05, 0.04] |
| **Asia** | [writer, consultant, translator, journalist, teacher] | [consultant, writer, journalist, lawyer, translator] |
| | [0.09, 0.06, 0.05, 0.05, 0.04] | [0.1, 0.09, 0.06, 0.04, 0.04] |
| **Europe** | [writer, consultant, journalist, nurse, teacher] | [writer, consultant, journalist, lawyer, producer] |
| | [0.1, 0.07, 0.05, 0.05, 0.04] | [0.11, 0.1, 0.06, 0.04, 0.04] |
| **Oceania** | [writer, consultant, teacher, nurse, journalist] | [writer, consultant, journalist, teacher, lawyer] |
| | [0.09, 0.07, 0.05, 0.04, 0.04] | [0.11, 0.08, 0.05, 0.04, 0.04] |

# I    Further Analysis for US Comparison

## I.1    Kendall's-Tau Coefficients

We use two quantitative measures of the relative deviation of GPT-2 predictions to US ground truth: mean-square error (MSE) (reported in Fig. 4 of the main paper) and Kendall-Tau coefficient (reported in Tab. 14). All Kendall-Tau coefficients signify a strong positive monotonous relationship between GPT-2's predictions and the US grouth truth, significant at the 1% level.

Table 14: GPT-2 vs US-data by gender share. Kendall-Tau ($K\tau$) coefficients of rank correlation.

| Intersection | $K\tau$ | p |
|---|---|---|
| Base | 0.628 | 0.000 |
| Asian | 0.428 | 0.001 |
| Black | 0.498 | 0.000 |
| Hispanic | 0.521 | 0.000 |
| White | 0.664 | 0.000 |

## I.2    Gender Predictions

Fig. 28 plots the percentage of women for each occupation as predicted by GPT-2 and as observed in the US Labor Bureau data. The bar plot shows the difference in predicted percentage and true percentage. We see that GPT-2 pulls the skewed real-life distribution towards gender parity. For example, GPT-2 predicts there to be more women mechanics, carpenters, taxi drivers, and police officers than there are in real life. Additionally, GPT-2 predicts there to be fewer women secretaries, maids, nurses, and models than observed in reality. Both of these examples suggest that GPT-2 under-predicts the number of women in heavily women-dominated jobs, and GPT-2 over-predicts the number of women in heavily men-dominated jobs. This supports our finding in the paper: although it may seem initially biased that GPT-2 predicts so many women to be secretaries and maids, the share of women within these occupations is actually higher in the US data.

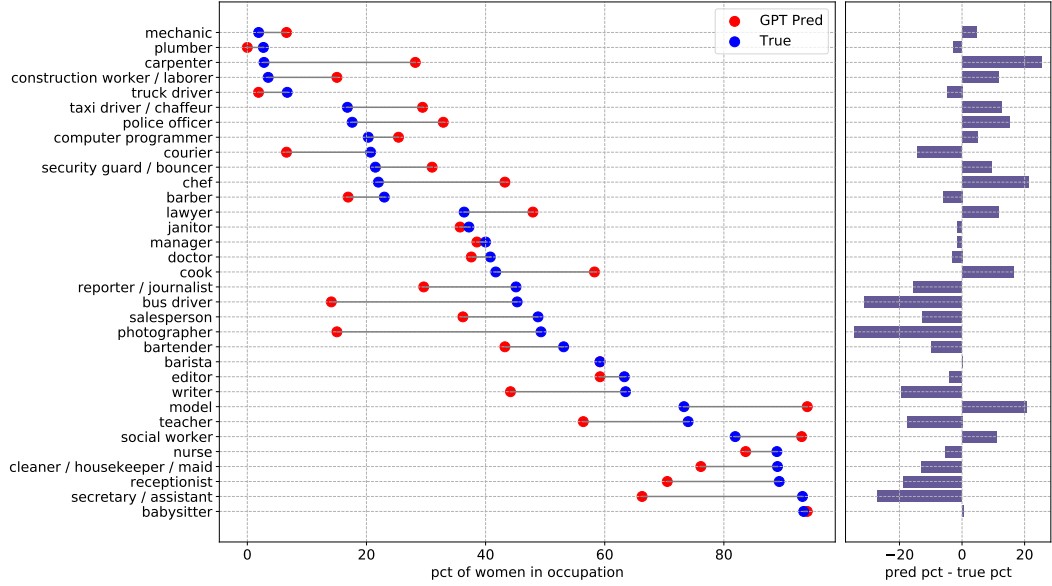

Figure 28: **GPT-2 predictions versus US data by gender share**. Difference in percentage of women predicted by GPT-2 and the percentage of women in the 2019 US Labor Bureau Statistics data, per occupation.

## I.3 Gender-Ethnicity Predictions

Fig. 29 presents the difference between US data and GPT-2's predicted proportions of gender-ethnicity pairs for the top 50 most frequently mentioned jobs matched with US occupational categories. The jobs on the y-axis are sorted by the true share of women in the US data. In line with the low mean-squared errors presented in the paper, GPT-2 accurately predicts the gender-ethnicity split for a given job, especially for Asian and Black workers. For jobs with a wide gender split, GPT-2 seems to corrects for societal skew. For example, it under-predicts the proportion of Hispanic women who are cleaners, housekeepers and maids by 34% (percentage points). Similarly, it under-predicts the proportion of Black men who are taxi drivers, chauffeurs or drivers, and the proportion of Hispanic men who are mechanics, plumbers, carpenters and construction workers. The proportion of White workers is less accurately predicted but the same pattern is observed towards under-predicting the proportion of women in female dominated jobs and over-predicting the proportion of women in male-dominated jobs.

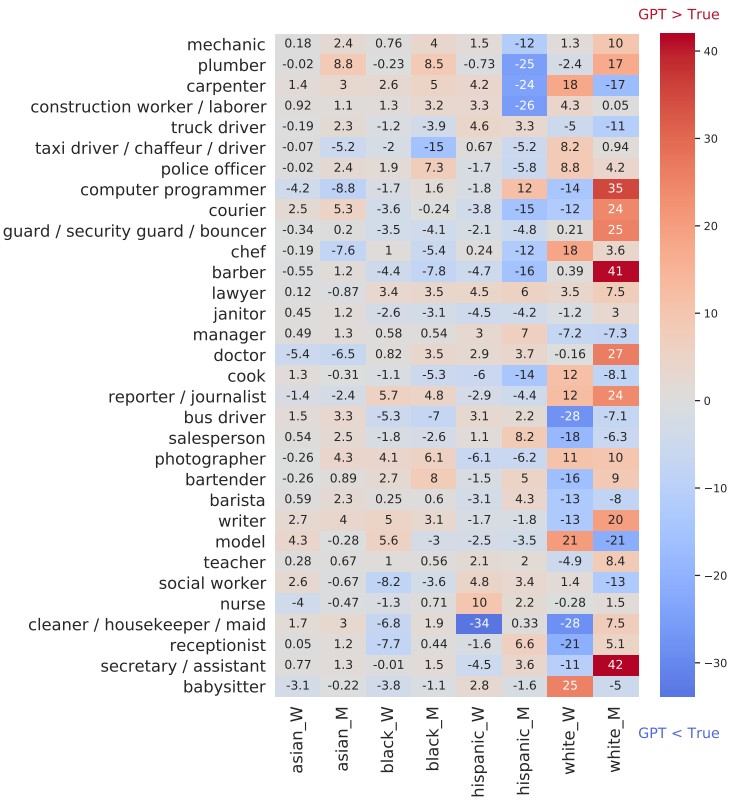

Figure 29: **GPT-2 predictions versus US data by gender-ethnicity intersection**. Red means that GPT-2 *over-predicts* the share of the occupation-ethnicity intersection pair; Blue means that GPT-2 *under-predicts* it.

# J   Companies Using AI for Hiring

Gartner has identified various use cases where AI can be useful in hiring process such as talent acquisition and HR virtual assistant (`https://www.gartner.com/en/newsroom/press-releases/ 2019-06-19-gartner-identifies-three-most-common-ai-use-cases-in-`). A number of companies are already using AI in hiring e.g. Aviro AI (`https://www. avrioai.com/features-and-benefits`) and Entelo (`https://www.entelo.com/ recruiting-automation/`). These companies have automated the hiring process and reducing human involvement in the job application assessment process. While this can have positive consequences, it can also have serious implications for people from marginalized groups if the occupational stereotypes and bias in the underlying AI models is not addressed.