# OpenReview forum: "Bias Out-of-the-Box: An Empirical Analysis of Intersectional Occupational Biases in Popular Generative Language Models"
_NeurIPS.cc/2021/Conference — NeurIPS 2021 Poster_

### Official Review · Reviewer_Gm9u · 2021-07-11

**Rating:** 6
**Confidence:** 4

**Summary:**

This paper focuses on probing GPT-2, as a text generative model, for the potential bias in its generated content. The authors used a template-based data collection approach (feeding GPT2 with pre-designed sentence parts and asking the GPT2 model to complete the sentence, using the default parameter). Particularly the authors presented results on:
1. Bias found in occupations between women and men
2. Intersectional interactions (such as religion + gender) have more significant effects on bias then just gender alone (shown by a series of logistic regression analysis)
3. Comparison with occupational statistics from US Labour Bureau showed that GPT2 even pulled back the effect of bias, considering the imbalance in occupational distribution in the real world.

While I appreciate the authors efforts to investigate bias (esp considering intersectional marginalized community), I recommend "a weak accept" of the paper in its present form, due to the lack of clarity in its presentation and several issues (please refer to weaknesses in the main review section). That said, I’d be willing to increase scores upon reading the authors’ rebuttal.


**Ethical Concerns:**

I commend the authors for their effort in analyzing potential bias inherent in a very popular model.

**Limitations And Societal Impact:**

The authors made great effort to acknowledge the limitations of their work.


**Main Review:**

Strengths:

1. The consideration of intersecting gender with other associations and the investigation of their effects are novel and extensively explored.
2. The authors collected a large number of data points, using GPT2-small, the most commonly used version, and therefore the results are particularly informative to many practitioners.


Weaknesses:

1. Comparing the occupational statistics computed by GPT2 vs those by the United States is very interesting and informative. However, the presentation on the methodology and the subsequent discussion is confusing to me. Particularly from section 3.4, I am not sure what “adj.” in equation (1) means and why “adj. Pred” is appropriate as a scaling factor. Would appreciate it if the authors could clarify and make this section clearer.
2. The analysis of intersection effects is interesting but I fail to see a clear presentation on statistical significance of these results. It may be clearer if the authors could specify p-values on some regressors and offer some discussions. From Table 3, I also do not believe that average pseudo-R2 is necessarily a meaningful measure for the individual factor.
3. The authors claim the contribution of “benchmarking the extent of bias relative to inherently skewed societal distributions of occupation associations”. However, I have some reservations as 1) the authors did not propose any quantitative measurement to the extent of occupation bias relative to real distributions in society; 2) the authors did not compare any models other than GPT2.
4. Several sections of the paper read confusing to me. There is a missing citation / reference in Line 99, section 3.1. The notation \hat{D}(c) from Line 165, section 3.4 is unreferenced.



**Time Spent Reviewing:**

4.5

---

> ### Author Response · Authors · 2021-08-10
> **Response to Reviewer Gm9u**
>
> ## Response to Reviewer Gm9u
>
> > While I appreciate the authors efforts to investigate bias (esp considering intersectional marginalized community), I recommend "a weak accept" of the paper in its present form, due to the lack of clarity in its presentation and several issues (please refer to weaknesses in the main review section). That said, I’d be willing to increase scores upon reading the authors’ rebuttal.
>
> Thank you for your detailed comments and feedback. In our response below, we address each of your comments in turn and provide further clarification. We hope these responses will improve your assessment of the paper and its contributions.
>
> > Particularly from section 3.4, I am not sure what “adj.” in equation (1) means and why “adj. Pred” is appropriate as a scaling factor. Would appreciate it if the authors could clarify and make this section clearer.
>
>
> In our paper, ‘adj.’ is an abbreviation for ‘adjusted’. We _adjust_ because our ‘artificial’ sample is evenly balanced (i.e., each gender and intersectional group has the same number of input-output sentences). The real-world distribution is not so evenly balanced for each gender and intersection group. Therefore, we adjust our ‘artificial’ shares of each intersectional group to more closely match those found in the US labor force. We will reformulate this paragraph to make this adjustment clearer.
>
> > The analysis of intersection effects is interesting but I fail to see a clear presentation on statistical significance of these results. It may be clearer if the authors could specify p-values on some regressors and offer some discussions. From Table 3, I also do not believe that average pseudo-$R^2$ is necessarily a meaningful measure for the individual factor.
>
> In the main paper we only had space to report aggregate significance because we ran 272 regressions. We thus focussed on comparing the significance of main effects (the gender dummy) relative to interaction effects (intersectional affiliations). However, in Appendix F.2 Fig. 10, we report the regression coefficients which were significant under the $p<0.05$ threshold. Additionally,  in Fig.11, we report the regression coefficient in each job regression. We will add the statistical results as a CSV table, in addition to the current presentation format, to the Supplementary Material.
>
> > The authors claim the contribution of “benchmarking the extent of bias relative to inherently skewed societal distributions of occupation associations”. However, I have some reservations as 1) the authors did not propose any quantitative measurement to the extent of occupation bias relative to real distributions in society; 2) the authors did not compare any models other than GPT2.
>
> To address the first point, we use mean-squared error (MSE) as a quantitative measurement of the relative deviation of GPT predictions to US Predictions (see Fig. 4). We will add the numbers from the legend in Fig. 4 to the main text, along with an explanation of why we chose this metric. We will provide further metrics to measure the differences in distribution, such as divergence measures and cross-entropy.
>
> To address the second point, we analyze XLNET (which was the second most downloaded model) for the baseline case (man vs woman) in Appendix D. In analyzing XLNET, we find similar patterns (for example, fewer jobs predicted for women) to those found in GPT-2. We will add a more in-depth comparison in the final version.
>
>
> > Several sections of the paper read confusing to me. There is a missing citation / reference in L.99, section 3.1. The notation $\hat{D}(c)$ from L.165, section 3.4 is unreferenced.
>
> We have fixed the missing link. It referred to a section in the Appendix (Appendix C). $\hat{D}(c)$ is in L.164 to define gamma. We have rewritten this sentence to make it clear $\hat{D}(c)$ is used in the scaling factor (gamma), then the scaling factor is used to define the adjusted prediction metric.

---

> > ### Comment · Reviewer_Gm9u · 2021-09-12
> > **Thank you for addressing the comment!**
> >
> > Thank you for addressing my comment.
> >
> > Overall, I have recommended accepting this paper. Other reviewers have listed venue and generalization issues. I advise the authors to address these issues particularly generalization issues (do you think such intersectional bias issues are prevalent in other models and why) in their future edits but I look forward to seeing this paper published.

---

### Official Review · Reviewer_BGEN · 2021-07-16

**Rating:** 7
**Confidence:** 5

**Summary:**

This paper tests occupational biases of GPT-2 (small) by generating many completions of "The X Y works as" (Y is either "man" or "woman", X is some ethnicity, religion, sexuality, or political affiliation word) and "Z works as a" where Z is a name associated with a gender and continent. The generations of GPT-2 are then compared against the US labor market (2019). Among other results, the paper finds that women are concentrated into a smaller number of professions, that some intersectional groups exacerbate this, and then generally the "X Y" prompt generates more skewed distributions than the "Z" prompt. Finally, the paper observes that the predicted probabilities roughly follow the US labor market, tending, in some cases, to "correct" for assymetries somewhat.


**Limitations And Societal Impact:**

The paper does an exemplary job of couching it's limitations. It briefly discusses societal impact.


**Main Review:**

Strengths:

1. The setup is very carefully done. The authors deserve great praise for thinking through details (eg. the COVID detail) and being very up front about assumptions and their consequences. I wish all papers would do as well as this one -- it's truly great to see!

2. The breadth of topics covered is very good, and the focus on intersectional effects is nice to see.

3. The analyses are done well, motivated well, and explained well.

4. The results give some insights into the behavior of these models.

Weaknesses:

1. I find myself missing what I should take away from this paper. It is a very nice study, and a very good reporting on the results of a study, but I don't know what to make of it. It's not surprising that GPT-2 encodes occupational stereotypes (we've seen similar results in many other models at this point) and, at least from eyeballing table 4, the intersectional effects seem fairly minor -- these lists look pretty consistent across each group, per GPT-2. It seems almost like GPT-2 is ignoring the race/ethnicity characteristic and just focusing on gender. Similar, in tables 2 and 3, the numbers seem really similar -- which are actually significant/meaningful? In contrast, there seems to be at least somewhat more difference in the actual US jobs proportions. The key question I'd like to see answered in the author response is: What are the take-aways from this study that we don't already know from prior work?

2. Similarly, the consistent result that women occupy fewer jobs according to GPT-2 seems moderately interesting on its face, but it's not clear to me (a) why this is happening or (b) why it matters. Is there, for instance, work in social science, that would explain this phenomenon? Is it true in the real world? Why is this good/bad, and what does it tell us about GPT-2 that's interesting to know? As a counterpoint, I could imagine that this effect doesn't have anything to do with bias in the model, but rather with bias in the US Labor coding. For instance, it could be the case that occupations that are historically occupied by women get lumped together into one profession (eg "nurse" covering many different aspects of the nursing profession) while occupations that are historically occupied by men get fine grained distinctions (eg assistant professor, associate professor, full professor, emeritus professor, affiliate professor, etc.). Any claim about the spread of occupations seems like it would be substantially impacted by this sort of coding issue (see, eg, Bowker and Star's classic book "Sorting Things Out"). (Note that I'm not saying that this is true, just that it's plausible and would need to be accounted for.)



Less significant comments:

 - while the inclusion of intersectional groups other than race/ethnicity is nice to see, because of the labor data the evaluation is really restricted to gender and race/ethnicity, which is unfortunate.

 - how is sampling done? how does it match with what other people do typically? (eg nucleus sampling, etc.)

 - The abstract states "downstream tasks" but generating text from a language model doesn't strike me as a "downstream task." I'm not sure what's meant here. This is also stated around l47.

 - line 36 "past month" -- which month is that?

 - In the data processing (l125), 10% and 20% of sentences are removed. Are these evenly distributed across categories or is there bias?

 - Why only one template?

 - Figure 2 is not readable in black and white

 - For some of the occupations that have inherant lexical gender (eg nun, waitress, etc.) it's not at all surprising to see gender skew. Why is this interesting?

 - The captions of the figures could be more informative -- for instance, figure 3 should state explicitly that this is from GPT-2.

 - The observations around line 224 could be from a defaulting effect. No one says "straight woman" then just say "woman" because straight is the default. This is actually a more general flaw with all of these prompt-based approaches, and the results here suggest it's meaningful.

 - around line 272, how does this interact with the data matching and filtering (namely where occupation names are matched)?

 - around line 302, this is a deeper question, but there is a big difference between "practicing a religion" and "being specifically called a Buddhist in text." That is, when people write text (which is used to train models like GPT-2) they have some communicative goal. They mention that someone is Buddhist if it is somewhat relevant for that communicative goal. Talking about someone's profession as a monk/nun would also plausibly be relevant to that communicative goal. So there's a big difference between "a person is Buddhist" and "a person is Buddhist and is explicitly mentioned as such in text." This of course confounds all results, but some discussion of this (see also Koller & Bender 2020) would help situate this work. Generally speaking I think the claim around religion is overstated.




**Time Spent Reviewing:**

3

---

> ### Author Response · Authors · 2021-08-10
> **Response to Reviewer BGEN (part 1)**
>
> ## Response to Reviewer BGEN
> We thank the reviewer for their detailed comments, encouraging words, and time spent on giving such thorough feedback! Below, we respond to the points and questions raised, and we illustrate how we have incorporate changes into the revised version of the paper.
>
> > I find myself missing what I should take away from this paper. It is a very nice study, and a very good reporting on the results of a study, but I don't know what to make of it…..Similar, in tables 2 and 3, the numbers seem really similar -- which are actually significant/meaningful? In contrast, there seems to be at least somewhat more difference in the actual US jobs proportions. The key question I'd like to see answered in the author response is: What are the take-aways from this study that we don't already know from prior work?
>
> Overall, there are four main takeaways from our work:
>   1. A protocol to quantify bias with regards to occupational associations conditioned on intersectional identities.
>   2. Empirical findings of unequal job diversity returned for men vs women (see next point's discussions).
>   3. Quantification of intersectional effects and their relative interplay with gender. Each intersection has a different magnitude of effect (e.g. see over-representation factor in Fig.5, Fig.6, and logistic regression results).
>   4. Comparison and discussion to real world data to assess 'relative bias' in language models versus US labor markets.
>
> Regarding Tab 2: Even though the variation in Gini-coefficients is small, it is clear that the variation is non-random due to the fact that all of the women intersections are grouped above all of the male intersections.
>
> Regarding Tab 3: We discuss the relative significance of the gender dummy versus the gender interactions in the text. The key takeaway here is that the gender dummy (main effect) is consistently a more significant determinant and contributes to a larger change in $R^2$ than the gender intersections (interaction effect).
>
>
> > The consistent result that women occupy fewer jobs according to GPT-2 seems moderately interesting on its face, but it's not clear to me (a) why this is happening or (b) why it matters. Is there, for instance, work in social science, that would explain this phenomenon? Is it true in the real world? Why is this good/bad, and what does it tell us about GPT-2 that's interesting to know?
>
> Thank you for the interesting insight on US labor coding, and for pointing us towards the  "Sorting Things Out" book (we have ordered it to read!). There is a body of work in labor market economics which explains occupational segregation and clustering. The pattern was commented on by Waldman and McEaddy (1974), who found women were clustered into fewer jobs than men. More recently, a 2014 report found a similar pattern: “Women are much more likely than men to be clustered in just a few occupations, with nearly half of all working women—44.4 percent—employed in just 20 occupations. Meanwhile, only about one-third—34.8 percent—of men are employed in the top 20 occupations for male workers” (Glynn, 2014).
>
> A wider body of research reveals why female-clustering in occupations is bad. Female-dominated industries have lower rates of pay than male-dominanted industries requiring similar levels of skills or education, and increasing clustering of women into few jobs leads to female work being devalued (England, 1992). In terms of mechanisms for why this happens in the real world, a number of factors have been proposed. As you mention, one reason is that women more commonly work in the domestic or informal sector (such as babysitter, housewife, full-time mother), so their work is less likely captured by official statistics. Other reasons for occupational segregation include flexibility of hours, part-time work and career breaks (Barbulescu and Bidwell, 2013; Grönlund and Magnusson, 2016); educational constraints (Borghans and Groot, 1993); and discrimination or stereotyping of female skills into ‘female-suited’ jobs (Beller, 1982).
>
> Relevant to the last of these mechanisms, we find GPT-2 overpredicts occupational clustering for the top five jobs returned for women as compared to the true clustering present in the US labor force (see L.282). This is true even if we hold the US labor coding bias fixed (i.e., comparing the same categories predicted by GPT-2 to the same categories in the US data). In terms of propagating damaging and self-fulfilling stereotypes over ‘female-suited’ jobs, we see this as a problematic form of bias in a widely-used language model.
>
> In terms of why GPT-2 might be overpredicting occupational clustering, Zhao et al (2018) report that, in the “OntoNotes” dataset, “male gendered mentions are more than twice as likely to contain a job title as female mentions. Moreover, these trends hold across genres". This dataset includes news and web data, which are similar types of sources to those on which GPT-2 was trained.
>
> We will add a section into our discussion on possible explanations for the bias observed in GPT-2 predictions of occupational variety. Here, we will summarise these points, including the insightful comment on bias in the US labor data coding.
>
>
> ### References
> Barbulescu, R., & Bidwell, M. (2013). Do women choose different jobs from men? Mechanisms of application segregation in the market for managerial workers. Organization Science, 24(3), 737–756. https://doi.org/10.1287/ORSC.1120.0757
>
> Beller, A. H. (1982). Occupational Segregation by Sex: Determinants and Changes. The Journal of Human Resources, 17(3), 371. https://doi.org/10.2307/145586
>
> Bender, E. M., & Koller, A. (2020, July). Climbing towards NLU: On meaning, form, and understanding in the age of data. In Proceedings of the 58th Annual Meeting of the Association for Computational Linguistics (pp. 5185-5198).
>
> Borghans, L., & Groot, L. (1993). Educational presorting and occupational segregation. Retrieved from www.elsevier.nlrlocatereconbase
>
> England, P. (Ed.). (1992). Comparable Worth: Theories and Evidence (1st ed.). Routledge. https://doi.org/10.4324/9781315080857
>
> Grönlund, A., & Magnusson, C. (2016). Family-friendly policies and women’s wages – is there a trade-off? Skill investments, occupational segregation and the gender pay gap in Germany, Sweden and the UK. http://Dx.Doi.Org/10.1080/14616696.2015.1124904, 18(1), 91–113. https://doi.org/10.1080/14616696.2015.1124904
>
> Glynn, S. J. (2014). Explaining the Gender Wage Gap. Retrieved from https://www.americanprogress.org/issues/economy/reports/2014/05/19/90039/explaining-the-gender-wage-gap/
>
> Waldman, E., & McEaddy, B. J. (1974). Where women work—an analysis by industry and occupation. Monthly Labor Review, 97(5), 3–13. Retrieved from https://www.jstor.org/stable/41839334?seq=1#metadata_info_tab_contents
>
> Sheng, E., Chang, K. W., Natarajan, P., & Peng, N. (2019). The woman worked as a babysitter: On biases in language generation. arXiv preprint arXiv:1909.01326.
>
> Sheng, E., Chang, K. W., Natarajan, P., & Peng, N. (2021). Societal Biases in Language Generation: Progress and Challenges. arXiv preprint arXiv:2105.04054.
>
>
> Solaiman, I., Brundage, M., Clark, J., Askell, A., Herbert-Voss, A., Wu, J., ... & Wang, J. (2019). Release strategies and the social impacts of language models. arXiv preprint arXiv:1908.09203.
>
> Zhao, J., Wang, T., Yatskar, M., Ordonez, V., & Chang, K.-W. (2018). Gender Bias in Coreference Resolution: Evaluation and Debiasing Methods. Retrieved from https://www.bls.gov/cps/cpsaat11.htm

---

> > ### Author Response · Authors · 2021-08-10
> > **Response to Reviewer BGEN (part 2)**
> >
> > ### Responses to “less significant comments”:
> >
> > > how is sampling done? how does it match with what other people do typically? (eg nucleus sampling, etc.)
> >
> > As discussed in the main paper on L.97 and Appendix C, we use top-$k$ sampling with the default values of top-$k=50$ and temperature=1.0. We analyse sample diversity as a function of these parameters as shown in Table 7 in Appendix C. Many works related to ours do not elaborate on these parameters, but Solaiman et al. (2019) use top-$k=40$ and temperature=1, which is similar.  Sampling methods do have an effect, and top-$k$ sampling is relatively high on the bias illuminating side of the spectrum (Sheng et al. 2021).
> >
> > > The abstract states "downstream tasks" but generating text from a language model doesn't strike me as a "downstream task." I'm not sure what's meant here. This is also stated around l47.
> >
> > Thank you for pointing this out. We have added a sentence in our introduction which more clearly explains the types of potential downstream tasks such as chatbots, virtual career advisors, unsupervised scanning of CV/resumes and text summarization.
> > > line 36 "past month" -- which month is that?
> >
> > We have amended this in our paper to refer to May 2021.
> >
> > > In the data processing (L.125), 10% and 20% of sentences are removed. Are these evenly distributed across categories or is there bias?
> >
> > The sentences generated using the ‘identity-based’ templates which were removed are evenly distributed across intersectional affiliations. Similarly, the sentences removed for the ‘name-based’ template are evenly distributed across continental origin and gender of the name. We will add these breakdowns to the appendix in a histogram.
> >
> >
> > > Why only one template?
> >
> > The template has several interchangeable parts which we vary. This allows us to analyse and isolate the effect of gender and its intersections. While also changing the template syntax would be interesting to examine the effect of prompt engineering, in this study we focus on one template and vary demographic details within this template, holding syntax fixed.
> >
> > > Figure 2 is not readable in black and white
> >
> > We have changed the ‘red’ woman lines to dashed lines so it can be viewed in color or black and white.
> >
> >
> >
> >
> > > For some of the occupations that have inherent lexical gender (eg nun, waitress, etc.) it's not at all surprising to see gender skew. Why is this interesting?
> >
> > We agree that GPT-2 is much more likely to predict occupations with an inherent lexical gender for women. This in itself is not interesting until we compare the predictions to US data. We make a comparison for ‘demographic distribution per occupation’ and find differences in the job tokens returned for men and women. But we also make a comparison for ‘occupational distribution per demographic’ where we conclude in L.321: “GPT-2's bias is not in the jobs associated with women, but in the rate at which it associates women with such a small set of jobs, a pattern exacerbated from the ground truth occupation data.” We discuss specific findings supporting this conclusion (see L.274): “While GPT-2 predicts 18% of Hispanic women to be waitresses, in reality only 3% of Hispanic women in America work as waitresses. Some of this strong association may be because ‘waitress’ is an inherently gendered job."
> >
> > Furthermore, while inherent lexical gender will affect GPT-2’s predictions, the majority of returned jobs are gender neutral. For example, as shown in Fig. 3, out of the 23 ‘female-dominated’ jobs shown, only two have explicit inherent lexical gender (waitress and housewife).
> >
> > > The captions of the figures could be more informative -- for instance, figure 3 should state explicitly that this is from GPT-2.
> >
> > We have amended the captions to be more explicit.
> >
> >
> > > The observations around line 224 could be from a defaulting effect. No one says "straight woman" then just say "woman" because straight is the default. This is actually a more general flaw with all of these prompt-based approaches, and the results here suggest it's meaningful.
> >
> > We agree this is an interesting point. To test this effect, we compare ‘woman’ with ‘straight woman’ or ‘lesbian woman’. We agree that ‘woman’ is likely defaulted to straight women but we additionally ask: what is the effect of explicitly mentioning her sexuality? We do find a difference in jobs returned for woman vs straight woman suggesting the explicit demarcation of sexuality has an effect.
> >
> > >  around line 272, how does this interact with the data matching and filtering (namely where occupation names are matched)?
> >
> > Generally, matching was required because the jobs returned by GPT-2 did not exactly align with categories in the US data, see Appendix E.3 (Tab. 11). The format of the US data is such that each job is not inherently gendered e.g. there is a category for 'waitress/waiters' in one row. The columns of the US data then give the proportion of women in that job, and the proportion of each ethnicity.  ‘Waitress’ and ‘waiter’ was a job returned by GPT-2. The US-labor data had a category for waitress/waiter so no complex matching or filtering occurred. However, if the taken statistic is in the 'Hispanic Women’ column of US labour data then we assume this is the number of ‘waitresses’ not 'waiters'. We will make this process clearer by adding detail to Appendix E.3.
> >
> > > around line 302, this is a deeper question, but there is a big difference between "practicing a religion" and "being specifically called a Buddhist in text." [...] This of course confounds all results, but some discussion of this (see also Koller & Bender 2020) would help situate this work. Generally speaking I think the claim around religion is overstated.
> >
> > Thank you for pointing us towards Koller & Bender (2020). We find their discussion of communicative intent particularly relevant to our work, and agree that the use of the explicit term ‘Buddhist man' as a linguistic choice likely conditions the returned set of jobs. As a comparison to this effect, we did include an additional name-origin template which tests the set of jobs returned when GPT-2 is not explicitly primed with an intersectional affiliation (see L.122). We do indeed find that these templates are more balanced by gender and by continental name origin (see L.248). To provide a better discussion of the explicit vs implicit demarcation of intersectionality, we will add more nuance to the points around religion in our discussion and include relevant points on communicative intent of language from Koller & Bender (2020).

---

> > ### Comment · Reviewer_BGEN · 2021-08-11
> > **Thank you for the very detailed response.**
> >
> > This is a very useful response and it's clear that the authors are quite familiar with the related literature. Thank you also for the clarifications and the parts that I missed in my original read of the paper.
> >
> > Overall I find that this paper is exceptionally well executed. I still think it could do a somewhat better job contrasting itself with previous literature (of which there is a lot [which the authors cite]) and so I hope some of this response makes it into the introduction, specifically saying what confirms previously known things vs what presents new findings.
> >
> > I've upped by score by one point.

---

### Official Review · Reviewer_FmSE · 2021-07-16

**Rating:** 5
**Confidence:** 4

**Summary:**

This work presents an empirical study of GPT-2 regarding the biases in the generated texts. Based on the generated texts, this work compares the gender and ethnicity distribution with the US Labour Bureau data and found that the distributions in the generated distributions are skewed, which demonstrates the biases embedded in the GPT-2 model.


**Limitations And Societal Impact:**

Some limitations are addressed in this paper.

**Main Review:**

- My first concern about this method is the text generation approach employed when collecting the 396K sentences. How to make sure the data generation process can reflect the intrinsic bias in the GPT-2? In other words, how do we know the template-based method did not misguide the model and then either amplify or reduce the bias in GPT-2?
- Regarding the selection of models, I agree GPT-2 is one of the popular models that has been used for text generation ever since it was released. However, it is unclear which observations are generic and which are only applied to GPT-2. I would like to argue a clear distinction between these two categories.
- It is an interesting idea to compare with the US Labor Market Data. However, from the perspective of research, I am not how rigorous this method is. For example, how do we know that the generated texts describe real-world situations instead of fictional facts? Besides, even the underlying assumption of this work hold and the data from both sides are consistent, that does not mean the GPT-2 does not have gender bias --- the bias in language is more subtle than some statistics

Overall, I think this work needs more justification on model selection and methodology selection.

**Time Spent Reviewing:**

3

---

> ### Author Response · Authors · 2021-08-10
> **Response to Reviewer FmSE**
>
> ## Response to Reviewer FmSE
> Thank you for your time and constructive feedback. In the following response, we address your questions and comments in detail, and describe how we will incorporate changes into our revised version.
>
>
> > My first concern about this method is the text generation approach employed when collecting the 396K sentences. How to make sure the data generation process can reflect the intrinsic bias in the GPT-2? In other words, how do we know the template-based method did not misguide the model and then either amplify or reduce the bias in GPT-2?
>
>
> We agree that measurement of bias will likely depend on the data generation process utilized. However, in this paper, we focus on one (very) specific kind of bias: that of intersectional biases with regards to occupations. This motivates our template-based method, which has been used in a similar form in previous works for analyzing biases (Sheng et al., 2019). In addition, we used this generation process to conduct thorough quantitative analyses based on repeated sampling of the same template, which we believe is especially important in this domain for reproducibility and robustness. For additional details on the method we selected, please see our first response to Reviewer Enfk.
>
>
> >Regarding the selection of models, I agree GPT-2 is one of the popular models that has been used for text generation ever since it was released. However, it is unclear which observations are generic and which are only applied to GPT-2. I would like to argue a clear distinction between these two categories.
>
> While comparing the biases across many models is out of scope for this paper, we agree that this is a very interesting point. We have already conducted an analysis in Appendix D using the XLNET model (second most downloaded model), and find similar patterns at least for the woman/man base case. We will further enhance this section by conducting the same intersectional analysis for XLNET as for GPT-2. We will include this in the revised Appendix D of the paper.
>
>
> > How do we know that the generated texts describe real-world situations instead of fictional facts? Besides, even the underlying assumption of this work hold and the data from both sides are consistent, that does not mean the GPT-2 does not have gender bias --- the bias in language is more subtle than some statistics
>
> While the text that GPT-2 generates is “fiction”, in that GPT-2 ‘invents’ text not directly based on factual events, the training corpora includes genres such as news sites and social media data, so its training is more heavily weighted towards real-world data and not towards fictional data such as the examples in the Book Corpus (Zhu et al., 2015).
>
> Propagating stereotypical job associations can be societally detrimental and cause associational harm, whether or not they are learnt from ‘fiction’ or ‘real-world situations’. We do not claim GPT-2 has no bias, only that its predicted occupational distribution is more balanced by gender relative to US labour market data.
>
> While it is true that bias in language can be subtle and its measurement is a non-trivial task, we focus on one specific domain and methodology in an attempt to quantify bias through repeated sampling. Quantifiable metrics are valuable even if they are imperfect. Crucially, we conduct a relative comparison between different identity attributes. Please also refer to our reply to Reviewer Enfk’s first point on the reasoning behind our measurement methodology.
>
>
> > Overall, I think this work needs more justification on model selection and methodology selection.
>
> We are happy to provide more details and explanations. Any pointers to what should be specifically added would be appreciated. For example, the other reviewers described the methods section as follows:
>
> * `Enfk`: “Justifications for certain methodological choices (like 2019 labor data, proportional scaling, US-centricity) are astutely included and the authors are transparent with limitations of the paper.”
> * `Bgen`: “The setup is very carefully done. The authors deserve great praise for thinking through details [...]  being very up front about assumptions and their consequences.”
>
>
>
> ### References
> Sheng, E., Chang, K. W., Natarajan, P., & Peng, N. (2019). The woman worked as a babysitter: On biases in language generation. arXiv preprint arXiv:1909.01326.
>
> Zhu, Y., Kiros, R., Zemel, R., Salakhutdinov, R., Urtasun, R., Torralba, A., & Fidler, S. (2015). Aligning books and movies: Towards story-like visual explanations by watching movies and reading books. In Proceedings of the IEEE international conference on computer vision (pp. 19-27).

---

### Official Review · Reviewer_Enfk · 2021-07-18

**Rating:** 7
**Confidence:** 4

**Summary:**

This paper analyzes the intersectional occupational bias in GPT-2. The authors propose a data collection protocol involving the generation of sentences using GPT-2 using identity-based and name-based templates, and the subsequent extraction of occupations using named entity recognition. Bias is analyzed with respect to gender, intersected with ethnicity, religion, sexuality, political affiliation and continental name origin. By comparing the distribution of predictions with US labor market distributions, the authors find that GPT-2 reflects, and sometimes corrects, the skewed societal distributions found in US labour market data.

The paper's contributions are the following:
- A bias probing and statistical analysis protocol
- Analyzing more categories for intersectional bias: ethnicity, religion, sexuality, political affiliation and continental name origin
- As far as I am aware, a new comparison of model predicted distributions with real-world societal distributions

**Limitations And Societal Impact:**

Yes


**Main Review:**

**Originality**

The question of examining bias in pre-trained language models is not new. However, the authors extend the use of prefix templates for conditional language generation (Sheng et al., 2019) by including more categories, and including a statistical analysis. The comparison of pre-trained langauge model predictions to real-world societal distributions is also a new contribution to the community.

I am chiefly concerned that the paper does not compare its proposed bias probing and statistical analysis protocol with existing methods (eg. embedding association tests, coreference resolution). While the authors do recognize this body of work exists and they cite them, I would have liked to see explanations of why their proposed protocol is better.

N.B. A comparison to sentiment classification in [21] is done and I agree with the authors, but is still insufficient to me.

- Can you evaluate your proposed bias detection protocol in relation to other methods (eg. embedding association tests)? Why do you think this is better?

**Quality**

The methods used are appropriate, and experimental results are well substantiated with comprehensive data and suitable illustrations. This paper is a complete work. Justifications for certain methodological choices (like 2019 labor data, proportional scaling, US-centricity) are astutely included and the authors are transparent with limitations of the paper.

- I am not convinced that one contribution of this work is to evaluate GPT-2 out-of-the-box. To my understanding, most other papers evaluating bias in large language models also do not tune hyperparameters before use, and the choice of model is often whichever was most salient at the time of experimentation.

**Clarity**

The paper is generally well written and structured. The provided appendix is also comprehensive.

- Line 99: missing link
- Line 166: should this be Pred(i, c)?
- Line 353: "going to made readily available"

**Significance**

Comparison with real-world societal distributions is an important contribution for the community, and will inform the debate on whether generative language models should reflect the real world or an ideal type world. In particular, the following results are useful:
- GPT-2 generated text indicates a greater clustering of women into fewer jobs
- Name origin has less of an effect on the occupation returned by GPT-2 than an explicit categorical intersection
- GPT-2 sometimes under-predicts the extent of occupational segregation, are interesting and useful to the community.

**Time Spent Reviewing:**

4

---

> ### Author Response · Authors · 2021-08-10
> **Response to Reviewer Enfk**
>
> ## Response to Reviewer Enfk
> Thank you for your detailed reading and understanding of our work, as well as for the encouraging feedback. In our response below, we address the points raised and describe how we have incorporated these into our revised version of the paper.
>
> > Can you evaluate your proposed bias detection protocol in relation to other methods (eg. embedding association tests)? Why do you think this is better?
>
> We chose the proposed protocol as it is well suited for probing generative language models in their most “natural” form, in which sentence completions are generated. In contrast to this, i) embedding association tests would require more heuristic choices, as they have been found to be highly dependent on the initial selection of seed words (Antoniak et al., 2021); ii) coreference resolution methods, such as Zhao et al. (2018), suffer from frequent ambiguities and unstated assumptions (Blodgett et al., 2021); and iii) information theoretic approaches, such as in Rudinger et al. (2017), require a pre-generated corpus and thus would confound the template-based generation with the bias measurement. We will add these points to our methodology section to give the reader a better understanding of related methods and our reasoning for choosing our method.
>
>
> > I am not convinced that one contribution of this work is to evaluate GPT-2 out-of-the-box. [...] the choice of model is often whichever was most salient at the time of experimentation.
>
>
> We agree that using a model out-of-the-box is not a novelty per se. Instead, it forms part of our method, as this way we can study GPT-2 how it is most likely to be applied (e.g. default parameters, most popular model version). This is also the reason why we have included the term “out-of-the-box” directly in the title. Additionally, in comparison to prior work, which as you mention choose models based on ‘saliency’, we aim to be fully transparent by specifying (1) _how_ we select the model (by popularity of downloads), (2) which _version_ of the model we use (gpt2-small);  and (3) the default parameter values (top-$k$ = 50, temperature = 1.0).
>
>
> ### Additional Points of Clarification
> * L.99: We have fixed the missing link - this linked to Appendix C.
> * L.166: We have fixed the typo.
> * L.353: We have fixed the typo - thank you for your keen eye!
>
>
> ### References
> Antoniak, M., & Mimno, D (2021). Bad Seeds: Evaluating Lexical Methods for Bias Measurement. In ACL.
>
> Blodgett, S. L., Lopez, G., Olteanu, A., Sim, R., & Wallach, H. (2021). Stereotyping Norwegian salmon: an inventory of pitfalls in fairness benchmark datasets. In Proc. 59th Annual Meeting of the Association for Computational Linguistics.
>
> Rudinger, R., May, C., & Van Durme, B. (2017, April). Social bias in elicited natural language inferences. In Proceedings of the First ACL Workshop on Ethics in Natural Language Processing (pp. 74-79).
>
> Zhao, J., Wang, T., Yatskar, M., Ordonez, V., & Chang, K. W. (2018). Gender bias in coreference resolution: Evaluation and debiasing methods. arXiv preprint arXiv:1804.06876.

---

> > ### Comment · Reviewer_Enfk · 2021-09-02
> > **Thank**
> >
> > Thank you for addressing my comments. My score remains at 7.

---

### Decision · Program_Chairs · 2021-09-27

**Decision:**

Accept (Poster)

**Comment:**

This paper investigates associations between occupations and demographic attributes in text generated by GPT-2 from a narrow range of templated prompts, and shows that GPT-2's generations indeed tend to mirror observed trends in the US labor market, opening up the potential for it to reinforce existing inequities.

Reviewers raised significant concerns about the novelty and significance of this work, but all agreed that the core claims are sound. During discussion, a consensus formed that it would be helpful to publish this work to help continue conversations about bias in self-supervised models at NeurIPS.